# Scattering amplitudes and BCFW in $\mathcal{N} = 2^*$ theory

Md. Abhishek[1,2⋆], Subramanya Hegde[1,2†], Dileep P. Jatkar[1,2‡] and Arnab Priya Saha[3∘]

**1** Harish-Chandra Research Institute, Chhatnag Road, Jhunsi, Allahabad, India-211019
**2** Homi Bhabha National Institute, Training School Complex,
Anushaktinagar, Mumbai, India-400094
**3** Indian Institute of Science Education and Research Bhopal, Bhopal Bypass Road,
Bhauri, Bhopal, India-462066

⋆ mdabhishek@hri.res.in , † subramanyahegde@hri.res.in ,
‡ dileep@hri.res.in , ∘ apsaha@iiserb.ac.in

## Abstract

We use massive spinor helicity formalism to study scattering amplitudes in $\mathcal{N} = 2^*$ super-Yang-Mills theory in four dimensions. We compute the amplitudes at an arbitrary point in the Coulomb branch of this theory. We compute amplitudes using projection from $\mathcal{N} = 4$ theory and write three point amplitudes in a convenient form using special kinematics. We then compute four point amplitudes by carrying out massive BCFW shift of the amplitudes. We find some of the shifted amplitudes have a pole at $z = \infty$. Taking the residue at $z = \infty$ into account ensures little group covariance of the final result.

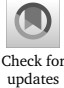
# 1 Introduction

The on-shell formulation of scattering amplitudes in quantum field theories has developed rapidly in the last couple of decades thanks to the clever use of the spinor helicity formalism for massless theories, see [1–11] and references therein. In particular, it has become a powerful tool in studying amplitudes in $\mathcal{N} = 4$ super-Yang-Mills(SYM) theories. It has also aided (super)gravity computations using the double copy formalism [12]. This formalism has also been extended to theories with lower supersymmetry, for some of the early works in this direction can be found in, *e.g.*, [13] and [14]. The spinor helicity formalism is well suited to study amplitudes in theories involving massless fields. However, for obvious reasons, it is important to extend this formalism to theories with massive fields, and there have been several steps taken in this direction already [15–22].

A natural extension is to do an excursion in the Coulomb branch of the $\mathcal{N} = 4$ SYM theory [23–25]. This is equivalent to studying amplitudes involving BPS states. Although BPS states are massless states from the higher dimensional point of view, they are massive states in four dimensions, and to accommodate them in the spinor helicity formalism one needs to double the number of spinor helicity variables. As mentioned earlier original spinor helicity variables are ideal for describing null momenta. The idea of doubling stems from the simple fact that any time-like momentum can be described in terms of two null momenta. Since each null momentum needs a pair of spinor helicity variables, we need doubling of the variables to describe the momenta of the massive fields. A related idea has also been explored earlier

where one utilises the fact that the long multiplets of $\mathcal{N}/2$-extended supersymmetry(SUSY) have the same number of states as the short multiplet of $\mathcal{N}$-extended SUSY algebra [17, 18].

The $\mathcal{N} = 2$ SYM theory is the next avenue to pursue in the increasing level of difficulty [13], but these theories have much richer structures and many interesting ones have non-vanishing $\beta$-function, see *e.g.*, [26, 27]. While it would be interesting to develop a general formalism that can encompass this kind of rich variety, it is often easier to look at the closest cousins of the model that is well understood. The $\mathcal{N} = 2^*$ theory is beautifully perched between the reasonably well understood $\mathcal{N} = 4$ theory and the wild variety of $\mathcal{N} = 2$ theories. The $\mathcal{N} = 2^*$ theory is therefore a natural meeting ground where one can test the generalised spinor helicity formalism before taking a plunge into studying amplitudes in the $\mathcal{N} = 2$ theory. With this motivation back in mind, in this paper, we address the problem of setting up an appropriate formalism for computing amplitudes in $\mathcal{N} = 2^*$ theory at an arbitrary point in the Coulomb branch. We explicitly compute three and four point amplitudes in the $\mathcal{N} = 2^*$ theory. The $\mathcal{N} = 2^*$ theory is obtained by writing $\mathcal{N} = 4$ SYM multiplet in terms of $\mathcal{N} = 2$ vector multiplet and $\mathcal{N} = 2$ adjoint hypermultiplet. If the adjoint hypermultiplet is massless then we get $\mathcal{N} = 4$ theory, but if the adjoint hypermultiplet is massive then it breaks $\mathcal{N} = 4$ SUSY down to $\mathcal{N} = 2$. The resulting theory with an $\mathcal{N} = 2$ vector multiplet coupled to massive adjoint hypermultiplet is referred to as the $\mathcal{N} = 2^*$ theory. This theory is in some sense a close relative of the $\mathcal{N} = 4$ Coulomb branch theory, in the sense that the techniques required to study the massive theory amplitudes are similar to those of the $\mathcal{N} = 4$ theory in the Coulomb branch. However, there is a crucial difference in the classical theory at the origin of the Coulomb branch of the $\mathcal{N} = 2^*$ theory we have massless vector multiplet coupled to a massive adjoint hypermultiplet, whereas in the Coulomb branch of the $\mathcal{N} = 4$ theory we recover massless $\mathcal{N} = 4$ SYM theory at the origin.

As emphasised above, one of our main motivations for studying amplitudes in the $\mathcal{N} = 2^*$ theory comes from the fact that we can use its connection with $\mathcal{N} = 4$ SYM to obtain useful lessons for amplitudes in $\mathcal{N} = 2$ theories. BCFW techniques for $\mathcal{N} = 2^*$ theory studied in this paper may be helpful for understanding recursion relations of amplitudes in $\mathcal{N} = 2$ theories. We employ two different techniques to compute the amplitudes in the $\mathcal{N} = 2^*$ theory. In section 2 we begin with the characterisation of $\mathcal{N} = 2$ multiplets, both massless and massive. We describe the massive multiplets in the $\mathcal{N}$-extended SUSY in terms of 'long' multiplets of $\mathcal{N}/2$ extended SUSY. We have put the word *long* in quotes because we employ the same technique for $\mathcal{N} = 2$ SUSY where the $\mathcal{N}/2$ extended SUSY is $\mathcal{N} = 1$ SUSY, which does not possess long multiplets. However, it does possess multiplets with respect to $SU(2)$ little group which helps organise the massive multiplets of $\mathcal{N} = 2$ SUSY. We also obtain $\mathcal{N} = 2$ massive multiplets by projection of $\mathcal{N} = 4$ multiplets, a method we use in the computation of three and four point amplitudes.

In section 3, we embark on the computation of three point amplitudes. After discussing the special kinematics for three point amplitudes of BPS states [7], we derive three point amplitudes using the method of projection from the $\mathcal{N} = 4$ theory. However, we find it convenient to write the expressions in terms of the $u$-spinor variable, that arises in three point special BPS kinematics, (see Eq.(31)) because this representation turns out to be suitable for carrying out the BCFW shift which is done in section 5. In the $\mathcal{N} = 2^*$ theory, we have only two types of three point amplitudes, one that involves three vectors or the other that involves one vector and two hypers. We derive both using the projection from $\mathcal{N} = 4$ theory. At the end of this section, we discuss the band structure of the scattering amplitudes. We note that, in three point amplitudes, besides the MHV and $\overline{\text{MHV}}$ bands that appear in the massless theory, the massive theory also has an MHV$\times\overline{\text{MHV}}$ band. In section 4 we compute four point function using the method of projection. In the $\mathcal{N} = 2^*$ theory there are only 3 types of four point amplitudes involving the massive vector as well as massive hyper, namely, a four massive vectors ampli-

tude, four massive hypers amplitude, and the one with two massive vectors and two massive hypers. We derive these amplitudes by taking appropriate projections.

The BCFW shift in section 5 for the $\mathcal{N} = 2^*$ theory does not follow from the $\mathcal{N} = 4$ theory by projection in a straightforward way because the shifts involved in the $\mathcal{N} = 4$ theory and those required to implement BCFW in the $\mathcal{N} = 2^*$ theory are different. In particular, the Grassmann variable $\eta_I^2$ which is shifted in the $\mathcal{N} = 4$ theory is projected out in the $\mathcal{N} = 2^*$ theory. As a result, the BCFW shifts are different in the two cases. In general, the massive BCFW shifts are not little group covariant. It is worth pointing out that this does not jeopardise the little group covariance of the final amplitude. This situation, in some sense, is analogous to the light cone gauge computations which do not maintain Lorentz covariance at every step but the final result is Lorentz covariant. The little group non-covariance manifests itself in the form of the integrand, as a function of the shift parameter $z$. One can therefore think of $z$ parametrising the little group non-covariance. In the $\mathcal{N} = 2^*$ case, we find that the amplitude containing gauge fields in the external legs do have a pole at $z = \infty$, and incorporating the residue from this pole is essential in getting the little group covariant answer for the amplitudes. We, in fact, find that the little group non-covariance is a blessing in disguise in the sense that the $z$ dependence of the integrand induced by it makes the non-covariant terms conspicuous. By accounting for the contribution of all the poles it is easy to see that the little group non-covariant contributions to the amplitude cancel pairwise and the final result is little group covariant. We turn this observation on its head to propose that the covariant amplitude can be obtained by simply ignoring the $z$ dependent parts of the integrand and hence ignoring the resulting pole at $z = \infty$. We believe this may be an efficient way of pulling out covariant expressions for amplitudes by leveraging the little group non-covariance. We end with concluding remarks in section 6 where we summarise our main results and speculate about the wider applicability of our procedure. Our notation and conventions as well as other technical details of some computations are relegated to appendices.

## 2 On-shell supermultiplets

In this section we will discuss the on-shell supermultiplets for $\mathcal{N} = 4$ and $\mathcal{N} = 2$ BPS multiplets. The BPS condition is defined in (108) in Appendix-A where we have listed the spinor helicity conventions relevant for this paper. Here we first recall the notation used for representation of BPS multiplets in $\mathcal{N} = 4$ super-Yang-Mills [18]. We then generalise it to construct BPS multiplets in $\mathcal{N} = 2$ theory. In the end we show that this same multiplet can be obtained by projection of the $\mathcal{N} = 4$ multiplet. We will use this method of projection in later sections.

### 2.1 $\mathcal{N} = 4$ SYM $1/2$-BPS multiplet

To set the stage for $\mathcal{N} = 2$ massive on-shell supermultiplets, let us first discuss the construction of $\mathcal{N} = 4$ SYM $1/2$-BPS multiplet [18]. We will utilise the supersymmetric massive spinor helicity formalism in four dimensions developed in [17]. The basic idea behind this construction is to capitalise the fact that the dimension of a short multiplet in the $\mathcal{N}$ extended supersymmetry is same as that of the long multiplet in the $\mathcal{N}/2$ extended supersymmetry. Therefore, to construct a $1/2$-BPS representation in $\mathcal{N}$-extended supersymmetry, one can use the long massive multiplets of $\mathcal{N}/2$ supersymmetry.

In the original $\mathcal{N}$-extended supersymmetry, a $1/2$-BPS representation has the same number of degrees of freedom as a massless representation. Therefore, when we take the massless limit of the $1/2$-BPS representation, it is merely a rearrangement of the components of the on-shell superfield for the massless representation. This rearrangement can be understood from the fact when we consider supersymmetry representations with the same maximum spin

or helicity, the Clifford vacua for the massive and for the massless theories are different. For instance, the helicity of the Clifford vacuum for the massless hypermultiplet is $h_0 = +1/2$ whereas the Clifford vacuum for massive hypermultiplet has spin $s_0 = 0$. We will see that this basis change is implemented by a half Fourier transform in the Grassmann variables that organise the on-shell supermultiplets.

The mapping between massless multiplets and massive 1/2-BPS multiplets can be implemented by using following steps.

- Represent the massive 1/2-BPS multiplet in $\mathcal{N}$-extended SUSY by using the massive long multiplet in the $\mathcal{N}/2$ SUSY.

- Take the massless limit of the superfield, by replacing the Grassmann variable $\eta_I^a$ (where $a$ is the $\mathcal{N}/2$ SUSY index and $I = \pm$ is the $SU(2)$ little group index) for massive fields by a new set variables $\eta^a, \tilde{\eta}^{\dagger a}$ using the rule $\eta_-^a \to \eta^a$ and $\eta_+^a \to \tilde{\eta}^{\dagger a}$.

- The massless $\mathcal{N}$ SUSY multiplet is then obtained by performing the half Fourier transform from $\tilde{\eta}^{\dagger a}$ to $\eta'^a$.

The set $(\eta^A = \eta^a, \eta'^a)$, are the appropriate Grassmann variables for the massless $\mathcal{N}$ SUSY superfield, and the half Fourier transform carried above achieves the necessary rearrangement of fields to change from the Clifford vacuum with helicity $s_0$ to Clifford vacuum with helicity $h_0$.

Let us now consider $\mathcal{N} = 4$ 1/2-BPS SYM multiplet. In [18], this was represented as a long massive vector multiplet[1] in $\mathcal{N} = 2$ supersymmetry which is given as,

$$\mathcal{W} = \phi + \eta_I^a \psi_a^I - \frac{1}{2}\eta_I^a \eta_J^b (\epsilon^{IJ}\phi_{(ab)} + \epsilon_{ab}W^{(IJ)}) + \frac{1}{3}\eta_I^b \eta_{Jb}\eta^{Ja}\tilde{\psi}_a^I + \eta_+^1 \eta_+^2 \eta_-^1 \eta_-^2 \tilde{\phi}. \tag{1}$$

To understand how one obtains the massless SYM multiplet in the massless limit of above, let us carry out the steps outlined earlier. By taking $\eta_-^a \to \eta^a$ and $\eta_+^a \to \tilde{\eta}^{\dagger a}$, we obtain,

$$\tilde{G} = \mathcal{W}|_{\eta_-^a \to \eta^a, \eta_+^a \to \tilde{\eta}^{\dagger a}}$$

$$= \phi + \eta^a \psi_a^- + \tilde{\eta}^{\dagger a}\psi_a^+ - \tilde{\eta}^{\dagger a}\eta^b \phi_{(ab)} - \tilde{\eta}^{\dagger a}\eta^b \epsilon_{ab}W^{(+-)} - \frac{1}{2}\eta^a \eta^b \epsilon_{ab}W^{(--)} - \frac{1}{2}\tilde{\eta}^{\dagger a}\tilde{\eta}^{\dagger b}\epsilon_{ab}W^{(++)}$$

$$+ \frac{2}{3}\tilde{\eta}_b^{\dagger b}\tilde{\eta}_b^{\dagger}\eta^a\tilde{\psi}_a^+ - \frac{2}{3}\eta^b \eta_b \tilde{\eta}^{\dagger a}\tilde{\psi}_a^- + \tilde{\eta}^{\dagger 1}\tilde{\eta}^{\dagger 2}\eta^1 \eta^2 \tilde{\phi}. \tag{2}$$

This representation is known as the non-chiral representation of the $\mathcal{N} = 4$ SYM multiplet [28]. To see this note that the helicity of the Clifford vacuum in the above superfield is $s_0 = 0$. However, we know that for the massless $\mathcal{N} = 4$ SYM representation theory, in the chiral representation, the helicity of the Clifford vacuum is $h_0 = 1$. To achieve this rearrangement of fields, let us implement a half Fourier transform of the Grassmann variables such that $\tilde{\eta}^{\dagger a}$ to $\eta'^a$. We get,

$$G = \int \prod_{a=1}^2 \left(d\tilde{\eta}^{\dagger a}e^{\tilde{\eta}^{\dagger a}\eta'^a}\right)\tilde{G}$$

$$= \eta'^1 \eta'^2 \phi + \eta^a \eta'^1 \eta'^2 \psi_a^- + \eta'^1 \psi_2^+ - \eta'^2 \psi_1^+ + \eta^1 \eta'^1 (\phi_{(12)} + W^{(+-)}) - \eta^2 \eta'^2 (\phi_{(12)} - W^{(+-)})$$

$$+ \eta^2 \eta'^1 \phi_{(22)} - \eta^1 \eta'^2 \phi_{(11)} + \eta^1 \eta^2 \eta'^1 \eta'^2 W^{(--)} - W^{(++)}$$

$$+ \frac{2}{3}\eta^a \tilde{\psi}_a^+ - \frac{2}{3}\eta^b \eta_b \eta'^1 \tilde{\psi}_2^- + \frac{2}{3}\eta^b \eta_b \eta'^2 \tilde{\psi}_1^- - \eta^1 \eta^2 \tilde{\phi}, \tag{3}$$

---

[1]We will interchangeably refer to the vector multiplet as the SYM multiplet in this paper.

where we used $\epsilon_{12} = -1$. The above superfield has helicity $h_0 = 1$ as expected. We can now rewrite the above by using $(\eta^A = \eta^a, \eta'^a)$ to obtain,

$$G = g^+ + \eta^A \lambda_A - \frac{1}{2}\eta^A \eta^B S_{AB} - \frac{1}{6}\eta^A \eta^B \eta^C \lambda^-_{ABC} - \eta^1 \eta^2 \eta^3 \eta^4 g^-, \tag{4}$$

where,

$$
\begin{aligned}
g^+ &= -W^{(++)}, & g^- &= -W^{(--)}, & S_{12} &= -\tilde{\phi}, & S_{34} &= -\phi, \\
S_{13} &= -W^{(+-)} - \phi_{(12)}, & S_{24} &= \phi_{(12)} - W^{(+-)}, & S_{14} &= -\phi_{(11)}, & S_{23} &= -\phi_{(22)}, \\
\lambda^-_{123} &= -\frac{4}{3}\tilde{\psi}^-_2, & \lambda^-_{234} &= -\psi^-_2, & \lambda^-_{134} &= -\psi^-_1, & \lambda^-_{124} &= -\frac{4}{3}\tilde{\psi}^-_1, \\
\lambda_1 &= \frac{2}{3}\tilde{\psi}^+_1, & \lambda_2 &= \frac{2}{3}\tilde{\psi}^+_2, & \lambda_3 &= \psi^+_2, & \lambda_4 &= -\psi^+_1. \tag{5}
\end{aligned}
$$

From the above, it becomes clear that the longitudinal mode of the massive W boson arises from the scalar fields of the massless SYM multiplet. Thus (1), describes the Coulomb branch of $\mathcal{N} = 4$ SYM. Note that even though the central charge structure is different for $\mathcal{N} = 4$ and $\mathcal{N} = 2$ supersymmetry, this is not relevant when considering on-shell representations. In this spirit, we will call massive multiplets in $\mathcal{N} = 1$ supersymmetry as long multiplets even though the central charge term is absent in $\mathcal{N} = 1$ supersymmetry. This nomenclature is helpful as massive $\mathcal{N} = 1$ multiplets can be used to describe 1/2-BPS $\mathcal{N} = 2$ multiplets analogous to how long $\mathcal{N} = 2$ multiplet can be used to represent 1/2-BPS $\mathcal{N} = 4$ SYM.

Before we proceed to construct $\mathcal{N} = 2$ supermultiplets using $\mathcal{N} = 1$ long multiplets, let us ask what is the massless limit of the $\mathcal{N} = 2$ SYM multiplet (1) within $\mathcal{N} = 2$ supersymmetry. To perform this massless limit, we need to take $\eta^a_+ \to \eta^a$ and $\eta^a_- \to \hat{\eta}^a$, where $\hat{\eta}^a$ organises the massless limit of the long massive supermultiplet in terms of distinct massless supermultiplets. This leads to,

$$
\begin{aligned}
\mathcal{W}\big|_{\eta^a_+ \to \eta^a, \eta^a_- \to \hat{\eta}^a} &= \phi + \eta^a \psi^-_a - \frac{1}{2}\eta^a \eta^b \epsilon_{ab} W^{(--)} + \hat{\eta}^a \psi^+_a - \hat{\eta}^a \eta^b \phi_{(ab)} - \hat{\eta}^a \eta^b \epsilon_{ab} W^{(+-)} \\
&\quad - \frac{2}{3}\hat{\eta}^a \eta^b \eta_b \tilde{\psi}^-_a - \frac{1}{2}\hat{\eta}^a \hat{\eta}_a W^{(++)} + \frac{2}{3}\hat{\eta}^a \hat{\eta}_a \eta^b \tilde{\psi}^+_b - \frac{1}{2}\hat{\eta}^a \hat{\eta}_a \eta^1 \eta^2 \tilde{\phi}. \tag{6}
\end{aligned}
$$

Therefore, we can write the above massless limit in terms of three massless superfields as,

$$\mathcal{W}\big|_{\eta^a_+ \to \eta^a, \eta^a_- \to \hat{\eta}^a} = \Phi + \hat{\eta}^a \Psi^+_a - \frac{1}{2}\hat{\eta}^a \hat{\eta}_a \mathcal{W}^{++}, \tag{7}$$

where the massless superfields are given as,

$$
\begin{aligned}
\Phi &= \phi + \eta^a \psi^-_a - \frac{1}{2}\eta^a \eta^b \epsilon_{ab} W^{(--)}, \\
\Psi^+_a &= \psi^+_a - \eta^b \phi_{(ab)} - \eta^b \epsilon_{ab} W^{(+-)} - \frac{2}{3}\eta^b \eta_b \tilde{\psi}^-_a, \\
\mathcal{W}^{++} &= W^{(++)} - \frac{4}{3}\eta^b \tilde{\psi}^+_b - \frac{1}{2}\eta^a \eta_a \tilde{\phi}. \tag{8}
\end{aligned}
$$

It is clear from the above that $\Phi$ and $\mathcal{W}^{(++)}$ superfields describe an $\mathcal{N} = 2$ SYM multiplet constructed in [14], and $\Psi^+_a$ describes a massless hypermultiplet. Notice that the longitudinal mode of the W boson in the long $\mathcal{N} = 2$ multiplet originates from the massless hypermultiplet. Therefore if we use (1) to describe a massive $\mathcal{N} = 2$ theory, then we are likely to obtain the Higgs branch of $\mathcal{N} = 2$ SYM. Notice that the on-shell hypermultiplet here occurs as a superfield which is a doublet under $R$-symmetry with Clifford vacuum of helicity $h_0 = 1/2$ being a doublet of fermions. This choice is appropriate as this organises the scalars in the hypermultiplet into

a triplet and a singlet under $R$-symmetry. This singlet is in fact the longitudinal mode of the $W$ boson in the massive theory. Therefore, the organisation of $R$-symmetry is consistent with the Higgs branch. However, in this paper, we are interested in Coulomb branch amplitudes as we are considering $\mathcal{N} = 2^*$ theory where the absence of massless hypermultiplets implies there is no Higgs branch.

## 2.2 $\mathcal{N} = 2$ Supersymmetry $1/2$-BPS multiplets

To study the amplitudes in the Coulomb branch of $\mathcal{N} = 2^*$ theory, we need the $1/2$-BPS on-shell superfields for SYM as well as hypermultiplets. These representations are obtained by using $\mathcal{N} = 1$ long massive multiplets as we show below. As before, the fact that central charge structures are different for $\mathcal{N}$-extended and $\mathcal{N}/2$-extended supersymmetry is not relevant for studying the on-shell representations. In our case, the $\mathcal{N} = 1$ long multiplets are those that were introduced in [17]. Here, we will show that these reproduce the right massless limit when they are used to describe $\mathcal{N} = 2$ $1/2$-BPS multiplets.

### 2.2.1 $\mathcal{N} = 2$ hypermultiplet

We can now try to construct the $\mathcal{N} = 2$ $1/2$-BPS hypermultiplet by using the long $\mathcal{N} = 1$ chiral multiplet which is given as,

$$\Phi = \phi + \eta_I \chi^I - \frac{1}{2} \eta_I \eta^I \tilde{\phi} \, . \tag{9}$$

Let us first implement $\eta_- \to \eta$ and $\eta_+ \to \tilde{\eta}^\dagger$. We then get,

$$\Phi = \phi + \eta \chi^- + \tilde{\eta}^\dagger \chi^+ + \tilde{\eta}^\dagger \eta \tilde{\phi} \, . \tag{10}$$

Half Fourier transform from $\tilde{\eta}^\dagger$ to $\eta'$ leads to,

$$\begin{aligned}
\tilde{\Phi} &= \int d\tilde{\eta}^\dagger (1 + \tilde{\eta}^\dagger \eta')(\phi + \eta \chi^- + \tilde{\eta}^\dagger \chi^+ + \tilde{\eta}^\dagger \eta \tilde{\phi}) \\
&= \eta' \phi - \eta \eta' \chi^- + \chi^+ + \eta \tilde{\phi} \, .
\end{aligned} \tag{11}$$

We can now relabel the Grassmann variables as, $\eta^1 = \eta$, $\eta^2 = \eta'$. We then obtain,

$$\tilde{\Phi} = \chi^+ + \eta^A \phi_A - \eta^1 \eta^2 \chi^- \, , \tag{12}$$

where $\phi_A = (\tilde{\phi}, \phi)$. This clearly the $\mathcal{N} = 2$ massless hypermultiplet.

There is one subtlety here, which is that unlike the $\mathcal{N} = 4$ SYM multiplet, the $\mathcal{N} = 2$ hypermultiplet is not self conjugate due to $SU(2)$ representation theory. This is complemented by the fact in the original $\mathcal{N} = 1$ theory, if the fermion is Dirac then one needs an anti-superfield. Therefore $\tilde{\Phi}$ and its anti-superfield provide the massless $\mathcal{N} = 2$ hypermultiplet. Thus $\mathcal{N} = 2$ massive hypermultiplet is represented by $\Phi$ from (9) as well as an anti-chiral superfield $\bar{\Phi}$ with the same structure.

### 2.2.2 $\mathcal{N} = 2$ SYM $1/2$-BPS multiplet

To construct the $1/2$-BPS $\mathcal{N} = 2$ SYM multiplet, consider the $\mathcal{N} = 1$ massive SYM multiplet,

$$\mathcal{W}^I = \lambda^I + \eta^I H + \eta_J W^{(IJ)} - \frac{1}{2} \eta_J \eta^J \tilde{\lambda}^I \, . \tag{13}$$

We now have a doublet of on-shell superfields under the little group. Let us first consider the case of $\mathcal{W}^+$ superfield. In $\eta, \tilde{\eta}^\dagger$ variables it reads,

$$\mathcal{W}^+ = \lambda^+ + \eta(W^{(+-)} + H) + \tilde{\eta}^\dagger W^{(++)} - \eta\tilde{\eta}^\dagger\tilde{\lambda}^+ \,. \tag{14}$$

Half Fourier transform yields,

$$
\begin{aligned}
\tilde{\mathcal{W}}^+ &= \int d\tilde{\eta}^\dagger(1 + \tilde{\eta}^\dagger\eta')(\lambda^+ + \eta(W^{(+-)} + H) + \tilde{\eta}^\dagger W^{(++)} - \eta\tilde{\eta}^\dagger\tilde{\lambda}^+) \\
&= \eta'\lambda^+ - \eta\eta'(W^{(+-)} + H) + W^{(++)} + \eta\tilde{\lambda}^+ \,.
\end{aligned} \tag{15}
$$

Finally in the $\eta^A = (\eta, \eta')$ variables,

$$\tilde{\mathcal{W}}^+ = g^+ + \eta^A\lambda_A - \eta^1\eta^2\varphi \,, \tag{16}$$

where $\lambda_A = (\tilde{\lambda}^+, \lambda^+)$, $g^+ = W^{(++)}$ and $\varphi = (W^{(+-)} + H)$. This is nothing but one of the on-shell superfields that represent massless $\mathcal{N} = 2$ SYM. Similarly, by considering the high energy limit of $\mathcal{W}^-$, we can recover the full $\mathcal{N} = 2$ massless SYM. We can see that the longitudinal component for the massive $W$ boson comes from the scalar in massless SYM. Therefore (13) is the $1/2$−BPS $\mathcal{N} = 2$ SYM on-shell superfield appropriate to describe the Coulomb branch of $\mathcal{N} = 2$ (as well as $\mathcal{N} = 2^*$) SYM.

## 2.3 Projection from $\mathcal{N} = 4$ supermultiplets to $\mathcal{N} = 2$ supersymmetry

We will now discuss a very useful connection between the $\mathcal{N} = 4$ SYM theory and a theory with $\mathcal{N} = 2$ SYM coupled to an adjoint $\mathcal{N} = 2$ hypermultiplet. For massless case, this connection has been discussed and utilised to present amplitudes for the $\mathcal{N} = 2$ theory in [29,30]. For the massive case, the relationship at the level of multiplets was discussed in the appendix of [18]. However, it was not utilised to write the amplitudes for $\mathcal{N} = 2^*$ theory. We will discuss the massless and the massive case below at the level of multiplets, which will help us write the amplitudes for $\mathcal{N} = 2^*$ theory in future sections.

### 2.3.1 Massless supermultiplet projection

Let us first consider the massless case. The massless on-shell superfield for $\mathcal{N} = 4$ SYM is given as,

$$G = g^+ + \eta^A\lambda_A - \frac{1}{2}\eta^A\eta^B S_{AB} - \frac{1}{6}\eta^A\eta^B\eta^C\lambda^-_{ABC} - \eta^1\eta^2\eta^3\eta^4 g^- \,. \tag{17}$$

If we expand the above on-shell superfield in the Grassmann variables $\eta^3$ and $\eta^4$, we obtain,

$$G_{\mathcal{N}=4} = G^+_{\mathcal{N}=2} + \eta^3\Phi_{\mathcal{N}=2} + \eta^4\bar{\Phi}_{\mathcal{N}=2} - \eta^3\eta^4 G^-_{\mathcal{N}=2} \,, \tag{18}$$

where,

$$
\begin{aligned}
G^+_{\mathcal{N}=2} &= g^+ + \eta^a\lambda_a - \eta^1\eta^2 S_{12} \,, \\
\Phi_{\mathcal{N}=2} &= \lambda_3 - \eta^a S_{3a} - \frac{1}{2}\eta^a\eta^b\lambda^-_{3ab} \,, \\
\bar{\Phi}_{\mathcal{N}=2} &= \lambda_4 - \eta^a S_{4a} - \frac{1}{2}\eta^a\eta^b\lambda^-_{4ab} \,, \\
G^-_{\mathcal{N}=2} &= S_{34} + \eta^a\lambda^-_{34a} - \eta^1\eta^2 g^- \,.
\end{aligned} \tag{19}
$$

These are the on-shell superfields for the $\mathcal{N}=2$ SYM and $\mathcal{N}=2$ hypermultiplet. To read off the amplitudes in the $\mathcal{N}=2$ theory, begin with the $\mathcal{N}=4$ SYM amplitude and expand in terms of $\eta^3$ and $\eta^4$ variables. If there is no $\eta^3$ or $\eta^4$ corresponding to a particular external leg then that leg corresponds to the superfield $G^+_{\mathcal{N}=2}$. Similarly, $\Phi_{\mathcal{N}=2}$ if there is only $\eta^3$, $\bar{\Phi}_{\mathcal{N}=2}$ if there is only $\eta^4$ and $G^-_{\mathcal{N}=2}$ if there is both $\eta^3, \eta^4$ corresponding to a particular external leg. Conversely, if one has the $\mathcal{N}=2$ superamplitudes, one can appropriately add them with factors of $\eta^3$ and $\eta^4$ to obtain the $\mathcal{N}=4$ superamplitude.

### 2.3.2 Massive supermultiplet projection

Since the massless projection discussed above proved useful in writing down the tree level amplitudes for $\mathcal{N}=2$ SYM coupled to a massless $\mathcal{N}=2$ hypermultiplet, it is natural to ask if such a connection exists between the massive 1/2-BPS multiplets. Let us now see how this can be done.

We had represented $\mathcal{N}=4$ SYM amplitude by using the $\mathcal{N}=2$ long multiplet given as,

$$\mathcal{W} = \phi + \eta_I^a \psi_a^I - \frac{1}{2}\eta_I^a \eta_J^b(\epsilon^{IJ}\phi_{(ab)} + \epsilon_{ab}W^{(IJ)}) + \frac{1}{3}\eta_I^b \eta_{Jb}\eta^{Ja}\tilde{\psi}_a^I + \eta_1^1\eta_1^2\eta_2^1\eta_2^2\tilde{\phi}\,. \tag{20}$$

Here, $a = 1, 2$. We can expand the above supermultiplet in terms of the Grassmann variables $\eta_I^2$ to obtain,

$$\mathcal{W} = \Phi + \eta_I^2 \mathcal{W}^I - \frac{1}{2}\eta_I^2\eta_J^2\epsilon^{IJ}\bar{\Phi}\,, \tag{21}$$

where,

$$\Phi = \phi + \eta_I^1 \psi_1^I - \frac{1}{2}\eta_I^1\eta_J^1\epsilon^{IJ}\phi_{11}\,,$$
$$\mathcal{W}^I = \psi_2^I - \eta^{1I}\phi_{12} + \eta_J^1 W^{(IJ)} - \frac{2}{3}\eta_J^1\eta^{J1}\tilde{\psi}_1^I\,,$$
$$\bar{\Phi} = \phi_{22} - \frac{2}{3}\eta_K^1\tilde{\psi}_2^K - \frac{1}{2}\eta_K^1\eta_L^1\epsilon^{KL}\tilde{\phi}\,. \tag{22}$$

Clearly, the above decomposition yielded long massive $\mathcal{N}=1$ chiral and anti-chiral as well as long-massive $\mathcal{N}=1$ SYM multiplet, which represent $\mathcal{N}=2$ massive hypermultiplet and $\mathcal{N}=2$ half-BPS SYM multiplet respectively.

Therefore, we propose that amplitude for $\mathcal{N}=2$ SYM in the Coulomb branch with a massive hypermultiplet can be obtained in non-chiral superspace by expanding the $\mathcal{N}=4$ SYM Coulomb branch amplitude in powers of $\eta_I^2$. No $\eta_I^2$ for a particular leg puts it in $\Phi_{\mathcal{N}=2}$, a single $\eta_I^2$ for a particular leg puts it in $\mathcal{W}^I$ and two $\eta_I^2$ for a particular leg puts it in $\bar{\Phi}_{\mathcal{N}=2}$.

## 3 Three point amplitudes

In this section, we will present the three point amplitudes for $\mathcal{N}=2^*$ theory. We will first review massless and massive three point special kinematics. We will then review the computation of massless three point amplitudes for $\mathcal{N}=2$ SYM coupled to an adjoint $\mathcal{N}=2$ hypermultiplet. Further, we will consider the three point amplitudes of $\mathcal{N}=4$ SYM to Coulomb branch to obtain three point amplitudes for $\mathcal{N}=2^*$ theory by projection. We will also elucidate the equivalence between different forms of results obtained from the projection and we give the three point amplitudes in terms of special $u$ spinors that will prove particularly useful for BCFW analysis in the next section.

## 3.1 Three point special kinematics

Let us first review the well known massless three point special kinematics [7]. Consider three particles with momentum $p_1, p_2, p_3$ respectively. We will consider all external momenta to be outgoing. Therefore, momentum conservation reads,

$$p_1^\mu + p_2^\mu + p_3^\mu = 0 \,. \tag{23}$$

This leads us to,

$$
\begin{aligned}
p_1^2 &= (-p_2 - p_3)^2 = 2p_2 \cdot p_3 = 0 \,, \\
p_2^2 &= (-p_1 - p_3)^2 = 2p_1 \cdot p_3 = 0 \,, \\
p_3^2 &= (-p_1 - p_2)^2 = 2p_1 \cdot p_2 = 0 \,.
\end{aligned}
\tag{24}
$$

From massless spinor helicity formalism, we have,

$$2p_i \cdot p_j = \langle ij \rangle [ij] \,. \tag{25}$$

By using (24) and momentum conservation, we can see that we can have two consistent limits for three particle special kinematics. Either,

$$[12] = [23] = [31] = 0 \,, \tag{26}$$

or

$$\langle 12 \rangle = \langle 23 \rangle = \langle 31 \rangle = 0 \,. \tag{27}$$

This is of course made possible by considering momenta to be complex, as for real momenta angle and square spinors are related by conjugation. When (26) is imposed, one obtains amplitudes written only in terms of angle spinors and vice versa for (27).

For massive particles, of interest to us is the three point special kinematics involving amplitudes for BPS and anti-BPS multiplets. The BPS condition along with central charge conservation for the amplitude will lead to a condition on the masses of the external legs. If we consider a three particle amplitude with two BPS and one anti-BPS multiplet then this condition will read $m_1 + m_3 = m_2$ where the second leg is taken to be anti-BPS. When this condition is satisfied, the following relation is satisfied.

$$
\begin{aligned}
-2p_1 \cdot p_2 + 2m_1 m_2 &= -p_3^2 - m_3^2 = 0 \,, \\
-2p_2 \cdot p_3 + 2m_2 m_3 &= -p_1^2 - m_1^2 = 0 \,, \\
2p_3 \cdot p_1 + 2m_3 m_1 &= p_2^2 + m_2^2 = 0 \,.
\end{aligned}
\tag{28}
$$

From massive spinor helicity formalism, we have,

$$
\begin{aligned}
-2p_1 \cdot p_2 + 2m_1 m_2 &= \frac{1}{2}([1^I 2^J] - \langle 1^I 2^J \rangle)([1^I 2^J] - \langle 1^I 2^J \rangle) \\
&= \det([1^I 2^J] - \langle 1^I 2^J \rangle) \,, \\
-2p_2 \cdot p_3 + 2m_2 m_3 &= \det([2^J 3^K] - \langle 2^J 3^K \rangle) \,, \\
2p_3 \cdot p_1 + 2m_3 m_1 &= \det([3^K 1^I] + \langle 3^K 1^I \rangle) \,.
\end{aligned}
\tag{29}
$$

Therefore, from (28), one obtains,

$$\det([i^I j^J] \pm \langle i^I j^J \rangle) = 0 \,, \tag{30}$$

where the relative minus sign occurs when one of the legs is BPS and the other is anti-BPS.

From (30), we see that the matrix $[i^I j^J] \pm \langle i^I j^J \rangle$ is of rank 1. Therefore we can write,

$$[i^I j^J] \pm \langle i^I j^J \rangle = u_i^I v_j^J, \tag{31}$$

where $i < j$ in cyclic ordering. Not all of these equations are independent. For instance,

$$([1^I 2^J] - \langle 1^I 2^J \rangle)([2_J 3^K] - \langle 2_J 3^K \rangle) = u_1^I v_2^J u_{2J} v_3^K. \tag{32}$$

The left hand side can be shown to vanish by using spin sums (101) and the expression for massive momenta in terms of massive spinors. This gives us,

$$v_2^I \propto u_2^I. \tag{33}$$

Similarly, we can see that all the $v$ spinors are proportional to the corresponding $u$ spinors. This leaves a scaling freedom, which we can use to set $v_i^I = u_i^I$ so that,

$$[i^I j^J] \pm \langle i^I j^J \rangle = u_i^I u_j^J, \tag{34}$$

where $i < j$ in cyclic ordering and the relative sign is as explained before. By contracting the above with $u_{iI}$ and $u_{jJ}$, we see that the above equations are solved by,

$$u_{1I}|1^I\rangle = u_{2J}|2^J\rangle = u_{3K}|3^K\rangle \equiv |u\rangle,$$
$$u_{1I}|1^I] = -u_{2J}|2^J] = u_{3K}|3^K] \equiv |u]. \tag{35}$$

The above equations capture the three point special BPS kinematics for massive particles. Recall that we have,

$$p_i = |i^I\rangle[i_I|. \tag{36}$$

It will be useful to see how to decompose this to manifest three point special BPS kinematics. For $u_{iI}$, consider dual variables $w_{iJ}$ such that $u_{iI} w_i^I = \epsilon^{IJ} u_{iI} w_{iJ} = 1$. We can insert this in the expression for momentum to obtain,

$$
\begin{aligned}
p_i &= u_{iJ} w_i^J |i^I\rangle[i_I| \\
&= u_{iI} w_i^J |i^I\rangle[i_J| + w_i^J |i^I\rangle(u_{iJ}[i_I| - u_{iI}[i_J|) \\
&= -|u\rangle[i^J|w_{iJ} + w_{iI}|i^I\rangle[i^J|u_{iJ} \\
&= -|u\rangle[i^J|w_{iJ} \pm w_{iI}|i^I\rangle[u|, \tag{37}
\end{aligned}
$$

where the relative sign is minus for anti-BPS leg due to the definition of $|u]$ spinor in (35). Further, variables $\hat{w}_{iI} = |u_i|w_{iI}$ will be useful for some manipulations. The $u$ and $v$ variables defined above have been considered before in the context of four and higher dimensional three particle special kinematics in [3, 11, 18]. We will see that these $u$-spinors will be useful to represent the three point amplitudes in a convenient way to simplify BCFW computations. We will note a few relations that we will use later. From (35), we have,

$$p_i|u\rangle = \pm m_i |u], \tag{38}$$

where the minus sign applies for anti-BPS legs. The multiplicative super-charges for three point amplitude are,

$$
\begin{aligned}
\frac{1}{\sqrt{2}} Q^{\dagger a} &= -\eta_{1I}^a |1^I\rangle - \eta_{2I}^a |2^I\rangle - \eta_{3I}^a |3^I\rangle, \\
\frac{1}{\sqrt{2}} Q_{a+2} &= \eta_{1I}^a |1^I] - \eta_{2I}^a |2^I] + \eta_{3I}^a |3^I]. \tag{39}
\end{aligned}
$$

From (38), we see that,

$$[uQ_{a+2}] = \langle uQ^{\dagger a}\rangle . \tag{40}$$

This shows that in the three particle super amplitude the delta functions in the two multiplicative supercharges are not independent and one has to account for the above relation to obtain the right supercharge conserving delta function as discussed in [18].

## 3.2 Three point massless amplitudes by projection

In the previous section, we discussed how the massless $\mathcal{N} = 4$ SYM on-shell superfield can be decomposed in terms of Grassmann variables $\eta^3, \eta^4$ to yield on-shell superfields for massless $\mathcal{N} = 2$ SYM and $\mathcal{N} = 2$ adjoint hypermultiplet. We can perform the same expansion of the $\mathcal{N} = 4$ SYM tree level amplitude to obtain the scattering amplitude for $\mathcal{N} = 2$ SYM coupled with an adjoint hypermultiplet. We will now perform this for three point amplitudes. We want to emphasize that all the three point amplitudes considered in this section and higher point ones studied in later sections are color ordered. Massless fields live in the adjoint representation of the gauge group. Let us compute the 3-point amplitudes. We know that the 3-point MHV amplitude in $\mathcal{N} = 4$ SYM is given as,

$$\mathcal{A}_3^{\text{MHV}}(G_{\mathcal{N}=4}, G_{\mathcal{N}=4}, G_{\mathcal{N}=4}) = \frac{i\delta^{(8)}(Q)}{\langle 12\rangle\langle 23\rangle\langle 31\rangle} , \tag{41}$$

where,

$$\delta^{2N}(Q) = \prod_{A=1}^{N} \sum_{i<j} \eta_i^A \eta_j^A \langle ij\rangle. \tag{42}$$

We get the desired massless $\mathcal{N} = 2$ amplitudes to be,

$$\mathcal{A}_3(G_{\mathcal{N}=2}^-, G_{\mathcal{N}=2}^-, G_{\mathcal{N}=2}^+) = \frac{-i\langle 12\rangle}{\langle 23\rangle\langle 31\rangle}\delta^{(4)}(Q), \qquad \mathcal{A}_3(G_{\mathcal{N}=2}^-, G_{\mathcal{N}=2}^+, G_{\mathcal{N}=2}^-) = \frac{-i\langle 31\rangle}{\langle 12\rangle\langle 23\rangle}\delta^{(4)}(Q),$$

$$\mathcal{A}_3(G_{\mathcal{N}=2}^+, G_{\mathcal{N}=2}^-, G_{\mathcal{N}=2}^-) = \frac{-i\langle 23\rangle}{\langle 12\rangle\langle 31\rangle}\delta^{(4)}(Q), \qquad \mathcal{A}_3(G_{\mathcal{N}=2}^-, \Phi, \bar{\Phi}) = \frac{i}{\langle 23\rangle}\delta^{(4)}(Q),$$

$$\mathcal{A}_3(G_{\mathcal{N}=2}^-, \bar{\Phi}, \Phi) = \frac{-i}{\langle 23\rangle}\delta^{(4)}(Q), \qquad \mathcal{A}_3(\Phi, G_{\mathcal{N}=2}^-, \bar{\Phi}) = \frac{i}{\langle 31\rangle}\delta^{(4)}(Q),$$

$$\mathcal{A}_3(\bar{\Phi}, G_{\mathcal{N}=2}^-, \Phi) = \frac{-i}{\langle 31\rangle}\delta^{(4)}(Q), \qquad \mathcal{A}_3(\Phi, \bar{\Phi}, G_{\mathcal{N}=2}^-) = \frac{i}{\langle 12\rangle}\delta^{(4)}(Q),$$

$$\mathcal{A}_3(\bar{\Phi}, \Phi, G_{\mathcal{N}=2}^-) = \frac{-i}{\langle 12\rangle}\delta^{(4)}(Q). \tag{43}$$

Amplitudes for hypermultiplets interacting with $G_{\mathcal{N}=2}^+$ come from the anti-MHV amplitude in $\mathcal{N} = 4$ SYM.

$$\mathcal{A}_3^{\text{anti-MHV}}(G_{\mathcal{N}=4}, G_{\mathcal{N}=4}, G_{\mathcal{N}=4}) = \frac{i\delta^{(4)}([12]\eta_3 + [23]\eta_1 + [31]\eta_2)}{[12][23][31]} . \tag{44}$$

We get the desired massless $\mathcal{N}=2$ amplitudes to be,

$$\mathcal{A}_3(G^+_{\mathcal{N}=2}, G^+_{\mathcal{N}=2}, G^-_{\mathcal{N}=2}) = \frac{-i[12]}{[23][31]}\delta^{(2)}([12]\eta_3 + [23]\eta_1 + [31]\eta_2),$$

$$\mathcal{A}_3(G^+_{\mathcal{N}=2}, G^-_{\mathcal{N}=2}, G^+_{\mathcal{N}=2}) = \frac{-i[31]}{[12][23]}\delta^{(2)}([12]\eta_3 + [23]\eta_1 + [31]\eta_2),$$

$$\mathcal{A}_3(G^-_{\mathcal{N}=2}, G^+_{\mathcal{N}=2}, G^+_{\mathcal{N}=2}) = \frac{-i[23]}{[12][31]}\delta^{(2)}([12]\eta_3 + [23]\eta_1 + [31]\eta_2),$$

$$\mathcal{A}_3(G^+_{\mathcal{N}=2}, \Phi, \bar{\Phi}) = \frac{i}{[23]}\delta^{(2)}([12]\eta_3 + [23]\eta_1 + [31]\eta_2),$$

$$\mathcal{A}_3(G^+_{\mathcal{N}=2}, \bar{\Phi}, \Phi) = \frac{-i}{[23]}\delta^{(2)}([12]\eta_3 + [23]\eta_1 + [31]\eta_2),$$

$$\mathcal{A}_3(\Phi, G^+_{\mathcal{N}=2}, \bar{\Phi}) = \frac{i}{[31]}\delta^{(2)}([12]\eta_3 + [23]\eta_1 + [31]\eta_2),$$

$$\mathcal{A}_3(\bar{\Phi}, G^+_{\mathcal{N}=2}, \Phi) = \frac{-i}{[31]}\delta^{(2)}([12]\eta_3 + [23]\eta_1 + [31]\eta_2),$$

$$\mathcal{A}_3(\Phi, \bar{\Phi}, G^+_{\mathcal{N}=2}) = \frac{i}{[12]}\delta^{(2)}([12]\eta_3 + [23]\eta_1 + [31]\eta_2),$$

$$\mathcal{A}_3(\bar{\Phi}, \Phi, G^+_{\mathcal{N}=2}) = \frac{-i}{[12]}\delta^{(2)}([12]\eta_3 + [23]\eta_1 + [31]\eta_2). \tag{45}$$

Four point and higher point amplitudes can also be obtained in a similar fashion as explained in [29,30]. We will see how an analogous projection can be used to write three and higher point amplitudes in the Coulomb branch of $\mathcal{N}=2^*$ theory. The three point amplitudes considered here will be obtained as high energy limits of three point amplitudes in $\mathcal{N}=2^*$ theory.

## 3.3 Massive three point amplitudes of $\mathcal{N}=2^*$ theory by projection from $\mathcal{N}=4$ SYM

In this section, we will discuss how to get the massive three point amplitudes for the Coulomb branch of $\mathcal{N}=2^*$ theory from the $\mathcal{N}=4$ Coulomb branch amplitude by using projection. It is to be noted that we are dealing with color ordered amplitudes. To illustrate this let us consider the case where the gauge group in massless $\mathcal{N}=4$ SYM theory at the origin of the moduli space is broken into two gauge groups, say $U(N+M) \to U(N) \times U(M)$ when we move away from the origin due to the presence of massive fields. Massless superfields live in the adjoint representations of either of the two gauge groups, whereas massive superfields are bifundamentals of $U(N) \times U(M)$ - they live in the fundamental representation of one group and anti-fundamental of the other [24]. More generally the gauge group of the massless theory is broken to multiple sub-groups, $\prod_k U(N_k)$ and the color ordering continues to hold. For convenience, we suppress the color indices of the fields while writing the amplitudes. Central charge conservation for the superamplitude dictates that in a three point amplitude one should take at most two BPS (anti-BPS) multiplets and the other one to be anti-BPS (BPS). The expression of massive three point $\mathcal{N}=4$ SYM amplitude where the second leg is taken to be anti-BPS and the other two legs to be BPS is given as [18],

$$
\begin{aligned}
\mathcal{A}_3[\mathcal{W}_1, \bar{\mathcal{W}}_2, \mathcal{W}_3] &= \frac{1}{m_3^2 \langle q|p_1 p_3|q\rangle}\delta^{(4)}(\mathcal{Q}^{\dagger a})\delta^{(2)}(\langle q|p_3|\mathcal{Q}_{a+2}]) \\
&= \frac{1}{\langle q|p_1 p_3|q\rangle}\delta^{(4)}(\mathcal{Q}_{a+2})\delta^{(2)}(\langle q\mathcal{Q}^{\dagger a}\rangle), \tag{46}
\end{aligned}
$$

where, the $R$-symmetry index $a = \{1, 2\}$, and the central charge conservation condition translates to $m_1 + m_3 = m_2$. Even though momenta corresponding to the first and the third leg

are manifestly present in the overall factor for the above expression, using momentum conservation $p_1 + p_2 + p_3 = 0$, we can replace these momenta with other pairs of momenta. As we had discussed earlier, three particle special kinematics renders the supercharges along the special $u$ spinor directions dependent. Therefore the reference spinor $|q\rangle$ is introduced such that $\langle qu\rangle \neq 0$. This helps us extract the component of the supercharge in the direction orthogonal to $|u\rangle$. It can be shown that any component amplitude is independent of the reference spinor. Thus the reference spinor is useful to organise the component amplitudes into super-amplitudes. The supercharges for $\mathcal{N} = 4$ massive theory are,

$$
\begin{aligned}
\mathcal{Q}_{a+2} &= |1^I]\eta_{1,I}^a - |2^I]\eta_{2,I}^a + |3^I]\eta_{3,I}^a, \\
\mathcal{Q}^{\dagger a} &= -|1^I\rangle\eta_{1,I}^a - |2^I\rangle\eta_{2,I}^a - |3^I\rangle\eta_{3,I}^a.
\end{aligned}
\tag{47}
$$

We will substitute this in (46) and expand in terms of $\eta_I^2$ variables to obtain,

$$
\begin{aligned}
\delta^{(4)}\big(Q^{\dagger a}\big) &= \delta^{(2)}\big(Q^\dagger\big)\big(\langle 1^I 2^J\rangle\eta_{1,I}^2\eta_{2,J}^2 + \langle 1^I 3^J\rangle\eta_{1,I}^2\eta_{3,J}^2 + \langle 2^I 3^J\rangle\eta_{2,I}^2\eta_{3,J}^2 \\
&\quad + \frac{1}{2}m_1\epsilon^{IJ}\eta_{1,I}^2\eta_{1,J}^2 + \frac{1}{2}m_2\epsilon^{IJ}\eta_{2,I}^2\eta_{2,J}^2 + \frac{1}{2}m_3\epsilon^{IJ}\eta_{3,I}^2\eta_{3,J}^2\big), \\
\delta^{(4)}(\mathcal{Q}_{a+2}) &= \delta^{(2)}(Q)\big(-[1^I 2^J]\eta_{1,I}^2\eta_{2,J}^2 + [1^I 3^J]\eta_{1,I}^2\eta_{3,J}^2 - [2^I 3^J]\eta_{2,I}^2\eta_{3,J}^2 \\
&\quad - \frac{1}{2}m_1\epsilon^{IJ}\eta_{1,I}^2\eta_{1,J}^2 - \frac{1}{2}m_2\epsilon^{IJ}\eta_{2,I}^2\eta_{2,J}^2 - \frac{1}{2}m_3\epsilon^{IJ}\eta_{3,I}^2\eta_{3,J}^2\big), \\
\delta^{(2)}(\langle q|p_3|\mathcal{Q}_{a+2}]) &= \delta(\langle q|p_3|Q])\big(\langle q|p_3|1^N]\eta_{1,N}^2 - \langle q|p_3|2^N]\eta_{2,N}^2 + \langle q|p_3|3^N]\eta_{3,N}^2\big), \\
\delta^{(2)}\big(\langle q\mathcal{Q}^{\dagger a}\rangle\big) &= \delta\big(\langle qQ^\dagger\rangle\big)\big(-\langle q1^N\rangle\eta_{1,N}^2 - \langle q2^N\rangle\eta_{2,N}^2 - \langle q3^N\rangle\eta_{3,N}^2\big),
\end{aligned}
\tag{48}
$$

where we have defined the super charges, $\mathcal{Q}^{\dagger 1} \equiv Q^\dagger$ and $\mathcal{Q}_3 \equiv Q$ for $\mathcal{N} = 2$ supersymmetry. As discussed earlier, due to three point special BPS kinematics these supercharges are not independent but satisfy the relation $\langle uQ^\dagger\rangle = [uQ]$. Thus, analogous to the $\mathcal{N} = 4$ case, we can write the three point supercharge conserving delta functions in two equivalent forms, $\delta^{(2)}\big(Q^\dagger\big)\delta\left(\langle q|p_i|Q]\right) = m_i\delta^{(2)}(Q)\delta\left(\langle qQ^\dagger\rangle\right), \ \forall i = \{1, 2, 3\}$.

Equipped with the above decomposition of $\mathcal{N} = 4$ supercharge conserving delta functions, we can calculate different three point amplitudes in $\mathcal{N} = 2^*$ by using projection. These three point amplitudes will play the role of the seed amplitudes in the BCFW computations in the later sections. Analogous to the massless $\mathcal{N} = 2$ SYM coupled with adjoint hypermultiplet, in the $\mathcal{N} = 2^*$ theory we have two types of three point amplitudes, one with all massive $\mathcal{N} = 2$ SYM vector multiplets $A_3\big[\mathcal{W}_1^I, \bar{\mathcal{W}}_2^J, \mathcal{W}_3^K\big]$ and another is the $\mathcal{N} = 2$ SYM vector multiplet interacting with one BPS-anti-BPS pair of the $\mathcal{N} = 2$ hypermultiplet, $A_3\big[\mathcal{W}_1^I, \bar{\Phi}_2, \Phi_3\big]$.

### 3.3.1 Amplitudes with one $\mathcal{N} = 2$ vector multiplet and two hypermultiplets

The three point amplitude which reflects the coupling between $\mathcal{N} = 2$ SYM and $\mathcal{N} = 2$ hypermultiplet involves one massive SYM multiplet with one BPS anti-BPS pair of hypermultiplets. To obtain this amplitude let us project out either expression of (46) with respect to the appropriate $\eta_I^2$ variables following (21),

$$
\begin{aligned}
A_3\big[\mathcal{W}_1^I, \bar{\Phi}_2, \Phi_3\big] &= \left(\frac{\partial}{\partial\eta_{1,I}^2}\right)\left(\frac{1}{2}\frac{\partial}{\partial\eta_{3,J}^2}\frac{\partial}{\partial\eta_3^{2,J}}\right)\mathcal{A}_3[\mathcal{W}_1, \bar{\mathcal{W}}_2, \mathcal{W}_3]\bigg|_{\eta_{i,I}^2 \to 0} \\
&= -\left(\frac{\langle q|p_3|1^I] + m_3\langle q1^I\rangle}{\langle q|p_1 p_3|q\rangle}\right)\delta^{(2)}(Q)\delta(\langle qQ^\dagger\rangle).
\end{aligned}
\tag{49}
$$

Since the above answer seems to prefer the external momentum $p_3$ we can try to see if there is any symmetry when it is replace with the momentum $p_2$ for the anti-BPS multiplet. We can

replace $p_3 = -(p_1 + p_2)$ and $m_3 = m_2 - m_1$ to get,

$$A_3\left[\mathcal{W}_1^I, \bar{\Phi}_2, \Phi_3\right] = -\left(\frac{\langle q|p_2|1^I] - m_2\langle q1^I\rangle}{\langle q|p_1 p_2|q\rangle}\right)\delta^{(2)}(Q)\delta(\langle qQ^\dagger\rangle). \tag{50}$$

Thus the three point amplitude is symmetric under the replacement $p_3 \to p_2$, $m_3 \to -m_2$. The minus sign $(-)$ before the factor with mass term $m_2$ reflects the fact that the second hypermultiplet is anti-BPS. One of the key observations in this paper is that there is a way to represent this amplitude which will make the BCFW computations significantly simpler.

The amplitude(49) in terms of the $u$-spinors can be represented as follows,

$$A_3\left[\mathcal{W}_1^I, \bar{\Phi}_2, \Phi_3\right] = -\left(\frac{\langle uq\rangle\langle u1^I\rangle}{m_1\langle q|p_1 p_3|q\rangle}\right)\delta^{(2)}(Q)\delta(\langle qQ^\dagger\rangle). \tag{51}$$

We have shown in Appendix-B.1 how this can be derived. In BCFW analysis, we will use this form for the three point amplitude.

**Massless limits:**   In the origin of the moduli space for $\mathcal{N} = 2^*$ theory, a massless $\mathcal{N} = 2$ SYM multiplet is coupled to the massive adjoint hypermultiplet. We will also be concerned with the amplitude at the origin of the moduli space. Therefore we take the high energy limit of the above three point amplitude where the $\mathcal{N} = 2$ SYM multiplet is made massless. As discussed in the previous section, the supermultiplet $\mathcal{W}^I$ goes to two on-shell superfields $G^+$ and $G^-$ in the high energy limit corresponding to the choices $I = +, -$ for the supermultiplet. We have the following three point amplitude with one positive helicity massless SYM multiplet and one pair of massive hypermultiplets,

$$A_3\left(G_1^+, \bar{\Phi}_2, \Phi_3\right) = \frac{1}{\langle q1\rangle}\delta^{(2)}(Q)\delta\left(\langle qQ^\dagger\rangle\right). \tag{52}$$

Central charge conservation equation implies $m_2 = m_3 = m$ with $m_1 = 0$. By applying special kinematic conditions one can verify the identity, $\langle q|p_2|Q] = -m\langle qQ^\dagger\rangle$, which you have used to express delta functions. Similarly, with one negative helicity massless SYM, we obtain,

$$A_3\left(G_1^-, \bar{\Phi}_2, \Phi_3\right) = -\frac{1}{\langle q|p_3|1]}\delta^{(2)}\left(Q^\dagger\right)\delta\left(\langle q|p_3|Q]\right). \tag{53}$$

The above amplitudes will be useful to construct massive four-point amplitudes with massless interchange which we have given in Appendix-D.

### 3.3.2   Amplitudes involving only $\mathcal{N} = 2$ vector multiplet

We will now calculate the following three point massive SYM amplitude in the concerning $\mathcal{N} = 2^*$ theory. Starting from the first expression of the amplitude (46), and taking the projections with respect to the $\eta_I^2$ variables, we get,

$$A_3\left[\mathcal{W}_1^I, \bar{\mathcal{W}}_2^J, \mathcal{W}_3^K\right] = \left(\frac{\partial}{\partial \eta_{1,I}^2}\frac{\partial}{\partial \eta_{2,J}^2}\frac{\partial}{\partial \eta_{3,K}^2}\right)\mathcal{A}_3[\mathcal{W}_1, \bar{\mathcal{W}}_2, \mathcal{W}_3]\Bigg|_{\eta_{i,I}^2 \to 0} \tag{54}$$

$$= -\left(\frac{\langle 1^I 2^J\rangle\langle q|p_3|3^K] + \langle 1^I 3^K\rangle\langle q|p_3|2^J] + \langle 2^J 3^K\rangle\langle q|p_3|1^I]}{m_3\langle q|p_1 p_3|q\rangle}\right)\delta^{(2)}(Q)\delta(\langle qQ^\dagger\rangle).$$

Similarly, if we start from the second expression of (46) and by taking similar projections we get,

$$A_3\left[\mathcal{W}_1^I, \bar{\mathcal{W}}_2^J, \mathcal{W}_3^K\right] = -\left(\frac{[1^I 2^J]\langle q3^K\rangle + [1^I 3^K]\langle q2^J\rangle + [2^J 3^K]\langle q1^I\rangle}{\langle q|p_1 p_3|q\rangle}\right)\delta^{(2)}(Q)\delta(\langle qQ^\dagger\rangle). \tag{55}$$

Here we have used the relations between the delta functions, $\delta^{(2)}\left(Q^\dagger\right)\delta\left(\langle q|p_3|Q]\right) = m_3\delta^{(2)}(Q)\delta\left(\langle qQ^\dagger\rangle\right)$ that arises from three particle special kinematics. The expressions obtained above by using two different forms for the $\mathcal{N}=4$ SYM amplitude appear to be different even though they describe the same amplitude. However, they are indeed equal and it can be shown if we multiply and divide the second expression with $[\rho u]$, where $|\rho\rangle = \frac{p_3}{m_3}|q\rangle$ and use Schouten identity we will obtain the first form. The use of $|u]$ spinor here is to convert square spinors into angle spinors after the Schouten identity has been used. i.e., we use $[ui^I] = \pm\langle ui^I\rangle$, where the minus sign applies to the anti-BPS leg.

We can further express this amplitude in terms of the $u$-spinors, in the following way,

$$A_3\left[\mathcal{W}_1^I, \bar{\mathcal{W}}_2^J, \mathcal{W}_3^K\right] = -\left(\frac{\langle uq\rangle\langle u1^I\rangle\langle 2^J 3^K\rangle}{m_1 m_3} + \frac{\langle uq\rangle\langle u2^J\rangle\langle 1^I 3^K\rangle}{m_2 m_3}\right)\frac{\delta^{(2)}(Q)\delta(\langle qQ^\dagger\rangle)}{\langle q|p_1 p_3|q\rangle}.$$ (56)

In the next section, we will see how this will simplify the BCFW computation.

**Massless limit:** If we take the first multiplet to be massless where $\mathcal{W}^+ \to G^+$, the amplitude (54) becomes,

$$A_3\left[G_1^+, \bar{\mathcal{W}}_2^J, \mathcal{W}_3^K\right] = -\frac{\langle 2^J 3^K\rangle\langle q|p_3|1]}{\langle q|p_1 p_3|q\rangle}\delta^{(2)}(Q)\delta(\langle qQ^\dagger\rangle).$$ (57)

Similarly, with $\mathcal{W}^- \to G^-$, we have from (55),

$$A_3\left[G_1^-, \bar{\mathcal{W}}_2^J, \mathcal{W}_3^K\right] = -\frac{[2^J 3^K]}{[1|p_3|q\rangle}\delta^{(2)}(Q)\delta(\langle qQ^\dagger\rangle).$$ (58)

The above amplitudes play the role of the seed amplitudes while calculating BCFW recursions with massless exchanges in Appendix-D.

## 3.4 Band structure

It is well known that massless $\mathcal{N}=4$ SYM amplitude has MHV and $\overline{\text{MHV}}$ configurations, however, in the case of massive theory, there is an additional configuration analogous to MHV$\times \overline{\text{MHV}}$ which vanishes in the high energy limit. In general, various helicity sectors of massless amplitudes combine into single little group covariant forms and this is referred to as band structure. Band structures of $\mathcal{N}=4$ SYM amplitudes in the Coulomb branch have been studied in [24].

Here we show the band structures of three-point massive vector amplitude in $\mathcal{N}=2^*$ SYM theory. Let us consider three-point amplitude with the third state being massless. Therefore the BPS constraint implies $m_1 = m_2 = m$. The supercharges are given by

$$\begin{aligned} Q^{\dagger 1} &= -|1^I\rangle\eta_{1,I}^1 - |2^I\rangle\eta_{2,I}^1 + |3\rangle\eta_3^1, \\ Q_2 &= |1^I]\eta_{1,I}^1 - |2^I]\eta_{2,I}^1 + |3]\tilde{\eta}_3^{\dagger 1}. \end{aligned}$$ (59)

Using three-point kinematics we find

$$\langle 3Q^{\dagger 1}\rangle = \langle 3|\frac{p_1}{m}|Q_2].$$ (60)

Three-point amplitude has supercharge conserving delta functions given by

$$\begin{aligned} &\delta^{(2)}\left(Q^{\dagger 1}\right)\delta\left(\langle q|p_1|Q_2]\right) \\ &= \frac{1}{\langle q3\rangle}\delta\left(\langle qQ^{\dagger 1}\rangle\right)\delta\left(\langle 3Q^{\dagger 1}\rangle\right)\delta\left(\langle q|p_1|Q_2]\right) \\ &= \frac{1}{\langle q3\rangle}\delta\left(\langle qQ^{\dagger 1}\rangle\right)\delta\left(\langle q|p_1|Q_2]\right)\left(\langle 31^+\rangle\zeta_+ + \langle 31^-\rangle\zeta_-\right), \end{aligned}$$ (61)

where we define $\zeta_I := \frac{1}{m}\langle 1_I Q^{\dagger 1}\rangle$. In the high energy limit for MHV amplitude we have the scaling $[i^+ j^+] \sim \mathcal{O}\left(\frac{m^2}{E}\right)$, whereas for $\overline{\text{MHV}}$ amplitude the scaling is $\langle i^- j^-\rangle \sim \mathcal{O}\left(\frac{m^2}{E}\right)$ [17]. Therefore in the above equation, either of the two terms survives in the high energy limit depending on the helicity configuration.

$$
\begin{aligned}
\langle 31^+\rangle &= -\langle 3|\frac{p_1}{m_1}|1^+] \\
&= -x[31^+], \qquad \because \frac{p_1}{m}|3\rangle = x|3].
\end{aligned}
\tag{62}
$$

So the first term gives rise to $\overline{\text{MHV}}$ amplitude in the massless limit.

$$
\begin{aligned}
\langle 31^+\rangle\zeta_+ &= \langle 31^+\rangle\frac{1}{m}\left(-\langle 1^- 1^+\rangle\eta_{1+} - \langle 1^- 2^+\rangle\eta_{2+} + \mathcal{O}\left(m^2\right)\right) \\
&= -\left(\langle 31^+\rangle\eta_{1+} + \langle 32^+\rangle\eta_{2+} + \mathcal{O}(m)\right) \\
&= \langle 3|p_1|Q_2], \qquad \text{as } m \to 0.
\end{aligned}
\tag{63}
$$

In the second equality we have used the Schouten identity,

$$
\langle 31^+\rangle\langle 1^- 2^+\rangle = -\left(\langle 1^- 3\rangle\langle 1^+ 2^+\rangle + \langle 1^+ 1^-\rangle\langle 32^+\rangle\right).
$$

The term $\langle 1^+ 2^+\rangle \sim \mathcal{O}\left(\frac{m^2}{E}\right)$ and hence is subleading.

Similarly for the MHV case second term in Eq.(61) can be expressed as

$$
\begin{aligned}
\langle 31^-\rangle\zeta_- &= \langle 31^-\rangle\frac{1}{m}\left(\langle 1^+ 1^-\rangle\eta_{1-} + \langle 1^+ 2^-\rangle\eta_{2-} + \mathcal{O}\left(m^2\right)\right) \\
&= -\left(\langle 31^-\rangle\eta_{1-} + \langle 32^-\rangle\eta_{2-} + \mathcal{O}(m)\right) \\
&= \langle 3 Q^{\dagger 1}\rangle, \qquad \text{as } m \to 0.
\end{aligned}
\tag{64}
$$

To be consistent with notations in the previous sections we will ignore superscript and subscript on the super-charges when taking the massless limit. The full three-point massless amplitudes are obtained by taking into account the prefactors multiplying the delta functions along with appropriate limits of the band structures as discussed above.

## 4 Four point $\mathcal{N} = 2^*$ amplitudes projected from $\mathcal{N} = 4$ SYM amplitudes

In this section, we will calculate the tree level massive (color ordered) four-point amplitudes of the $\mathcal{N} = 2^*$ theory by using projection, similar to how we obtained three point amplitudes in the previous section. The four-point massive $\mathcal{N} = 4$ SYM amplitude with two BPS multiplet ($\mathcal{W}$) and two anti-BPS multiplet ($\bar{\mathcal{W}}$) is given by,

$$
\mathcal{A}_4[\mathcal{W}_1, \bar{\mathcal{W}}_2, \mathcal{W}_3, \bar{\mathcal{W}}_4] = \frac{\delta^{(4)}(Q^{\dagger a})\delta^{(4)}(Q_{a+2})}{s_{12}s_{41}},
\tag{65}
$$

where, the masses for the external legs satisfy central charge conservation relation $m_1 + m_3 = m_2 + m_4$. The generalized Mandelstam variables are defined as $s_{ij} = -(p_i + p_j)^2 - (m_i \pm m_j)^2$, and the four particle supercharges of the $\mathcal{N} = 4$ theory with $a = \{1, 2\}$ are given by,

$$
\begin{aligned}
Q^{\dagger a} &= -|1^I\rangle\eta_{1,I}^a - |2^I\rangle\eta_{2,I}^a - |3^I\rangle\eta_{3,I}^a - |4^I\rangle\eta_{4,I}^a, \\
Q_{a+2} &= |1^I]\eta_{1,I}^a - |2^I]\eta_{2,I}^a + |3^I]\eta_{3,I}^a - |4^I]\eta_{4,I}^a.
\end{aligned}
\tag{66}
$$

As evident from the notation above, the legs 1 and 3 are BPS and the rest are anti-BPS. We now decompose the above supercharges in terms of $\eta_I^2$ variables to get,

$$
\begin{aligned}
\delta^{(4)}(\mathcal{Q}^{\dagger a}) \;=\;& \delta^{(2)}(Q^\dagger)\Big( \langle 1^I 2^J \rangle \eta_{1,I}^2 \eta_{2,J}^2 + \langle 1^I 3^J \rangle \eta_{1,I}^2 \eta_{3,J}^2 + \langle 1^I 4^J \rangle \eta_{1,I}^2 \eta_{4,J}^2 \\
& + \langle 2^I 3^J \rangle \eta_{2,I}^2 \eta_{3,J}^2 + \langle 2^I 4^J \rangle \eta_{2,I}^2 \eta_{4,J}^2 + \langle 3^I 4^J \rangle \eta_{3,I}^2 \eta_{4,J}^2 \\
& + \frac{1}{2}\epsilon^{IJ}\Big[ m_1 \eta_{1,I}^2 \eta_{1,J}^2 + m_2 \eta_{2,I}^2 \eta_{2,J}^2 + m_3 \eta_{3,I}^2 \eta_{3,J}^2 + m_4 \eta_{4,I}^2 \eta_{4,J}^2 \Big]\Big), \\
\delta^{(4)}(\mathcal{Q}_{a+2}) \;=\;& \delta^{(2)}(Q)\Big( -[1^I 2^J]\eta_{1,I}^2 \eta_{2,J}^2 + [1^I 3^J]\eta_{1,I}^2 \eta_{3,J}^2 - [1^I 4^J]\eta_{1,I}^2 \eta_{4,J}^2 \\
& - [2^I 3^J]\eta_{2,I}^2 \eta_{3,J}^2 + [2^I 4^J]\eta_{2,I}^2 \eta_{4,J}^2 - [3^I 4^J]\eta_{3,I}^2 \eta_{4,J}^2 \\
& - \frac{1}{2}\epsilon^{IJ}\Big[ m_1 \eta_{1,I}^2 \eta_{1,J}^2 + m_2 \eta_{2,I}^2 \eta_{2,J}^2 + m_3 \eta_{3,I}^2 \eta_{3,J}^2 + m_4 \eta_{4,I}^2 \eta_{4,J}^2 \Big]\Big),
\end{aligned}
\tag{67}
$$

where $Q^\dagger$ and $Q$ are the super charges for the $\mathcal{N}=2^*$ theory as introduced in the previous section.

**a)**  Let us first calculate the massive 4-point amplitude in the $\mathcal{N}=2^*$ theory with two BPS hypermultiplets ($\Phi_{\mathcal{N}=2^*}$) and two anti-BPS hypermultiplets ($\bar{\Phi}_{\mathcal{N}=2^*}$) external legs, by taking the projection from Eq.(65) as,

$$
\begin{aligned}
A_4[\Phi_1, \bar{\Phi}_2, \Phi_3, \bar{\Phi}_4] \;=\;& \left( \frac{1}{2}\frac{\partial}{\partial \eta_{1,K}^2}\frac{\partial}{\partial \eta_1^{2K}} \right)\left( \frac{1}{2}\frac{\partial}{\partial \eta_3^{2K}}\frac{\partial}{\partial \eta_{3,N}^2} \right) \mathcal{A}_4[\mathcal{W}_1, \bar{\mathcal{W}}_2, \mathcal{W}_3,, \bar{\mathcal{W}}_4]\Big|_{\eta_{i,I}^2 \to 0} \\
\;=\;& \left( -\langle 1^I 3^J \rangle [1_I 3_J] - 2m_1 m_3 \right)\frac{\delta^{(2)}(Q^\dagger)\delta^{(2)}(Q)}{s_{12}s_{41}} \\
\;=\;& \frac{s_{13}}{s_{12}s_{41}}\delta^{(2)}(Q^\dagger)\delta^{(2)}(Q),
\end{aligned}
\tag{68}
$$

where we have used $s_{13} = -(p_1+p_3)^2 - (m_1+m_3)^2 = -2p_1 \cdot p_3 - 2m_1 m_3 = -\langle 1^I 3^J \rangle [1_I 3_J] - 2m_1 m_3$ which can be deduced with the help of relations given in Appendix-A.

**b)**  Similarly, we can calculate the 4-point $\mathcal{N}=2^*$ massive SYM amplitude with two BPS and anti-BPS combinations by projection,

$$
\begin{aligned}
A_4[\mathcal{W}_1^I, \bar{\mathcal{W}}_2^J, \mathcal{W}_3^K, \bar{\mathcal{W}}_4^L] =& \left( \frac{\partial}{\partial \eta_{1,I}^2}\frac{\partial}{\partial \eta_{2,J}^2}\frac{\partial}{\partial \eta_{3,K}^2}\frac{\partial}{\partial \eta_{4,L}^2} \right)\mathcal{A}_4[\mathcal{W}_1, \bar{\mathcal{W}}_2, \mathcal{W}_3,, \bar{\mathcal{W}}_4]\Big|_{\eta_{i,I}^2 \to 0} \\
=& -\Big( \langle 1^I 2^J \rangle[3^K 4^L] + \langle 1^I 3^K \rangle[2^J 4^L] + \langle 1^I 4^L \rangle[2^J 3^K] \\
& + \langle 2^J 3^K \rangle[1^I 4^L] + \langle 2^J 4^L \rangle[1^I 3^K] + \langle 3^K 4^L \rangle[1^I 2^J] \Big)\frac{\delta^{(2)}(Q^\dagger)\delta^{(2)}(Q)}{s_{12}s_{41}}.
\end{aligned}
\tag{69}
$$

**c)**  Finally, by a similar procedure, we can calculate the 4-point amplitude with two massive $\mathcal{N}=2^*$ SYM and two massive hypermultiplets,

$$
\begin{aligned}
A_4[\mathcal{W}_1^I, \bar{\mathcal{W}}_2^J, \Phi_3, \bar{\Phi}_4] =& \left( \frac{\partial}{\partial \eta_{1,I}^2}\frac{\partial}{\partial \eta_{2,J}^2} \right)\left( -\frac{1}{2}\epsilon_{KL}\frac{\partial}{\partial \eta_{3,K}^2}\frac{\partial}{\partial \eta_{3,L}^2} \right)\mathcal{A}_4[\mathcal{W}_1, \bar{\mathcal{W}}_2, \mathcal{W}_3,, \bar{\mathcal{W}}_4]\Big|_{\eta_{i,I}^2 \to 0} \\
=& \left( \frac{\langle 1^I|p_3|2^J] + \langle 2^J|p_3|1^I] - m_3\langle 1^I 2^J \rangle - m_3[1^I 2^J]}{s_{12}s_{41}} \right)\delta^{(2)}(Q^\dagger)\delta^{(2)}(Q)
\end{aligned}
$$

$$= -\left( \frac{\langle 1^I|p_4|2^J] + \langle 2^J|p_4|1^I] + m_4\langle 1^I 2^J\rangle + m_4[1^I 2^J]}{s_{12}s_{41}} \right) \delta^{(2)}(Q^\dagger)\delta^{(2)}(Q).$$

(70)

From the second last equality to the last one of the above calculation, we have shown that the final expression is symmetry under the interchange $p_3 \to p_4$ and $m_3 \to -m_4$. The negative signs indicate that the one hypermultiplet is BPS and the other is anti-BPS.

It is easy to show that the other 4-point massive $\mathcal{N} = 2^*$ amplitudes with different combinations of multiplets (for example, three hyper with one SYM) do not exist by counting the number of Grassmann variables.

The results obtained in the previous section and this section for three and four point amplitudes using projection is one of the main results of this paper as well as convenient expressions for three particle amplitudes in terms of $u$-spinors. In the next section, using these results as an anchor, we attempt to compute the four point amplitudes in $\mathcal{N} = 2^*$ amplitude by using supersymmetric massive BCFW. Even though the seed three point amplitudes and the final four point amplitudes are related to the $\mathcal{N} = 4$ results by projection, BCFW computation for $\mathcal{N} = 2^*$ are significantly different. This is because, when we carried out the projection, we factored out the Grassmann variables $\eta_I^2$ from both the three point as well as four point amplitudes. However, these variables also played a role in the BCFW analysis in $\mathcal{N} = 4$ theory. Due to this subtlety, we will see that unlike $\mathcal{N} = 4$, in $\mathcal{N} = 2^*$ we will encounter poles at infinity in the BCFW analysis. We will further see some nice features which are highlighted since the answer is known from the projection independently.

# 5  Four point amplitudes for $\mathcal{N} = 2^*$ theory from BCFW

In this section, we will construct the four point amplitudes from three point amplitudes obtained in section-3. Even though we obtained the four point amplitudes using projection in the previous section, BCFW analysis to obtain these amplitudes with the results from projection as anchor could give us insights on massive super BCFW in $\mathcal{N} = 2$ theories. This is precisely one of the main motivations to study $\mathcal{N} = 2^*$ theory as we can utilise its intimate connection with $\mathcal{N} = 4$ SYM to obtain general insights on $\mathcal{N} = 2$ theories.

## 5.1  Massive BCFW shifts

Massive BCFW has been discussed in [18, 21, 22, 31]. We will review the massive super-BCFW analysis from [18] and point out subtleties in $\mathcal{N} = 2^*$ Coulomb branch BCFW as compared to the Coulomb branch of $\mathcal{N} = 4$ SYM.

The massive BCFW shifts are necessarily little group non-covariant as the $SU(2)$ little group structure makes it impossible to write a covariant shift for two legs. The shift can be defined as follows. Consider a shift in $i$ and $j$ legs. The shift in momentum is,

$$\hat{p}_i = p_i + zr, \qquad \hat{p}_j = p_j - zr,$$

(71)

where $p_i \cdot r = p_j \cdot r = r \cdot r = 0$. We will use Mandelstam variables appropriate for massive scattering. i.e., $s_{kl} = -(p_k + p_l)^2 - (m_k \pm m_l)^2$, where the relative sign occurs if one leg is BPS and another is anti-BPS. Under the above shift a Mandelstam variable $s_{ik}$ where $k \neq i, j$ will be shifted as,

$$\hat{s}_{ik} = -s_{ik}\frac{(z - z_I)}{z_I}, \qquad z_I = \frac{s_{ik}}{2p_k \cdot r}.$$

(72)

To obtain $r$ which satisfies the required properties, it is useful to write momenta $p_i$ and $p_j$ in terms of two null momenta. This defines a special frame. The detailed properties of this special frame are not important for this paper. In this special frame, one can solve for the null momentum $r$ and it breaks the little group covariance. One can further supersymmetrise the above shift by finding shifts for the supercharges that respect the BPS condition. These shifts in supercharges also break little group covariance. However, since the final amplitude will be covariant, the little group non-covariance must cancel. In $\mathcal{N} = 4$ SYM Coulomb branch, it was found that the $z$ dependence of the four point amplitude drops out before performing the contour integral in $z$. Therefore, the precise forms of the shift were found to be irrelevant.

In $\mathcal{N} = 2^*$ theory Coulomb branch, even though the amplitudes are related to that of $\mathcal{N} = 4$ theory by projection as discussed in previous sections, the BCFW analysis is not identical. This happens because while applying projection we project out the $\eta_I^2$ Grassmann variables which were also shifted in the case of $\mathcal{N} = 4$ theory. This subtlety leads to the appearance of a pole at infinity ($z \to \infty$) in the BCFW analysis for four point amplitudes in $\mathcal{N} = 2^*$. However, we find that in the contour integral the integrand, obtained by gluing three-point amplitudes, furnishes the covariant expression of the desired four-point amplitude provided we ignore the $z$-dependent terms. $z$-dependent terms are non-covariant and there are mutual cancellations of the little group non-covariant terms coming from all the residues including the boundary term. This property may have validity beyond $\mathcal{N} = 2^*$ theory as we show in Appendix-D.1, where we consider scattering amplitude with massless $\mathcal{N} = 2$ hypermultiplet interchange. Even though the amplitude matches that of $\mathcal{N} = 2^*$ theory, this particular channel is absent in $\mathcal{N} = 2^*$ due to the absence of massless hypermultiplets.

## 5.2 Amplitude with two massive hypers and two massive vector multiplets

Let us consider the four-point amplitude $A_4\left[\mathcal{W}_1^I, \bar{\mathcal{W}}_2^J, \Phi_3, \bar{\Phi}_4\right]$ with two massive hyper and two massive SYM multiplet in the $\mathcal{N} = 2^*$ theory. Here, we take legs 1 and 3 to be BPS and the rest are anti-BPS, and all external states are taken to be outgoing. With these conventions, momentum conservation reads $p_1 + p_2 + p_3 + p_4 = 0$, and the central charge conservation relation is given as $m_1 + m_3 = m_2 + m_4$. We consider BCFW shifts in legs 3 and 4 with the complex parameter $z$ as,

$$\begin{aligned} \hat{p}_3 &= p_3 + zr\,, \\ \hat{p}_4 &= p_4 - zr\,, \end{aligned} \tag{73}$$

where the null momentum $r$ breaks little group covariance and satisfies orthogonality properties as discussed earlier. With this choice of shift only the $s_{14}$-channel diagram will contribute to the massive amplitude.

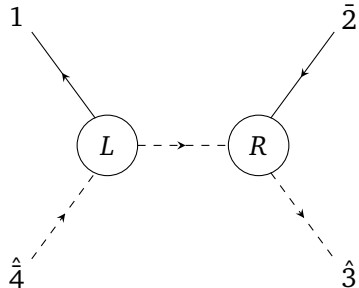

The incoming arrows indicate the outgoing anti-BPS states in order to show the central charge flow. After contour deformation away from $z = 0$, we can write,

$$A_4\left[\mathcal{W}_1^I, \bar{\mathcal{W}}_2^J, \Phi_3, \bar{\Phi}_4\right] = \oint_{z \neq 0} \frac{dz}{z} \frac{z_I}{z - z_I} \int d^2\eta_{\hat{P}} A_L\left[\mathcal{W}_1^I, \hat{\bar{\Phi}}_4, \Phi_{\hat{P}}\right] \frac{-1}{s_{14}} A_R\left[\bar{\Phi}_{-\hat{P}}, \hat{\Phi}_3, \bar{\mathcal{W}}_2^J\right], \tag{74}$$

where, the massive exchange momentum $\hat{P} = -(p_1 + \hat{p}_4)$. The generalized Mandelstam variable appeared in this diagram $s_{14} = -(p_1 + p_4)^2 - (m_1 - m_4)^2$ and the value of the simple pole $z_I = -\frac{s_{14}}{2r \cdot p_1}$. The left and right three-point amplitudes can be expressed in terms of the $u$-spinors as,

$$
\begin{aligned}
A_L\left[\mathcal{W}_1^I, \hat{\bar{\Phi}}_4, \Phi_{\hat{P}}\right] &= \frac{\langle u^{(L)} q\rangle \langle u^{(L)} 1^I\rangle}{m_1^2 \langle q|\hat{P} p_1|q\rangle} \delta^{(2)}\left(\hat{Q}_L^\dagger\right) \delta\left(\langle q|p_1|\hat{Q}_L]\right), \\
A_R\left[\bar{\Phi}_{-\hat{P}}, \hat{\bar{\Phi}}_3, \bar{\mathcal{W}}_2^J\right] &= -\frac{\langle u^{(R)} q\rangle \langle u^{(R)} 2^J\rangle}{m_2^2 \langle q|\hat{P} p_2|q\rangle} \delta^{(2)}\left(\hat{Q}_R^\dagger\right) \delta\left(\langle q|p_2|\hat{Q}_R]\right).
\end{aligned}
\tag{75}
$$

For the left amplitude, we take the hypermultiplet with $\hat{P}$ momentum to be outgoing BPS state with mass $m_P$. For the right amplitude the hypermultiplet with momentum $-\hat{P}$ to be outgoing anti-BPS state with mass $m_P$. Throughout the calculation, we have used the following analytic continuations of the massive spinors and the Grassmann variables.

$$
|-P^I] = i|P^I], \qquad |-P^I\rangle = i|P^I\rangle, \qquad u_{-P,I}^{(R)} = i u_{P,I}^{(R)}, \qquad \eta_{-P}^I = i\eta_P^I.
\tag{76}
$$

Clubbing the delta functions in left and right amplitudes, and performing the $\eta_{\hat{P}}$ integration, we obtain,

$$
\int \mathrm{d}^2\eta_{\hat{P}} \delta^{(2)}\left(\hat{Q}_L^\dagger\right) \delta\left(\langle q|p_1|\hat{Q}_L]\right) \delta^{(2)}\left(\hat{Q}_R^\dagger\right) \delta\left(\langle q|p_2|\hat{Q}_R]\right) = m_1 m_2 \frac{\langle u^{(R)} q\rangle \langle u^{(L)} q\rangle}{(u_{\hat{P},M}^{(R)} u_{\hat{P}}^{(L)M})} \delta^{(2)}(Q^\dagger) \delta^{(2)}(Q).
\tag{77}
$$

The detailed calculations are presented in Appendix-B.2 and we finally get,

$$
\begin{aligned}
&A_4\left[\mathcal{W}_1^I, \bar{\mathcal{W}}_2^J, \Phi_3, \bar{\Phi}_4\right] \\
&= \frac{1}{s_{14}} \oint_{z \neq 0} \frac{\mathrm{d}z}{z} \frac{z_I}{z - z_I} \frac{u_{1,K}^{(L)} \langle 1^K 1^I\rangle u_{2,L}^{(R)} \langle 2^L 2^J\rangle}{m_1 m_2} \frac{\left(u_{\hat{P}M}^{(R)} u_{\hat{P}}^{(L)M}\right)}{\left(u_{\hat{P}M}^{(R)} u_{\hat{P}}^{(L)M}\right)^2} \delta^{(2)}\left(Q^\dagger\right) \delta^{(2)}(Q) \\
&= -\oint_{z \neq 0} \frac{\mathrm{d}z}{z} \frac{z_I}{z - z_I} \left(\frac{\langle 1^I|\hat{p}_4|2^J] + \langle 2^J|\hat{p}_4|1^I] + m_4\langle 1^I 2^J\rangle + m_4[1^I 2^J]}{s_{12} s_{41}}\right) \delta^{(2)}(Q^\dagger) \delta^{(2)}(Q) \\
&= -\left(\frac{\langle 1^I|p_4|2^J] + [1^I|p_4|2^J\rangle + m_4[1^I 2^J] + m_4\langle 1^I 2^J\rangle + z_I\langle 1^I|r|2^J] + z_I[1^I|r|2^J\rangle + \mathcal{B}}{s_{12} s_{14}}\right) \delta^{(2)}(Q^\dagger) \delta^{(2)}(Q).
\end{aligned}
\tag{78}
$$

Here we have used, $\left(u_{\hat{P}M}^{(R)} u_{\hat{P}}^{(L)M}\right)^2 = -s_{12}$ which is explained in the appendix of [18], and we know, $\langle q|\hat{P} p_1|q\rangle = \langle u^{(L)} q\rangle^2$, also, $\langle q|\hat{P} p_2|q\rangle = \langle u^{(R)} q\rangle^2$. We emphasize that in the penultimate step of the above equation the integrand is nothing but the four-point amplitude that we obtained using projection from $\mathcal{N} = 4$ SYM amplitude, but with deformed momenta. This is consistent with the fact that $\oint_{z=0} \mathrm{d}z \frac{\mathcal{A}(z)}{z} = \mathcal{A}$ and one can simply read off the four-point amplitude from the integrand by ignoring the $z$-dependent terms.

To evaluate the boundary term, $\mathcal{B}$, we substitute $z = \frac{1}{u}$ and calculate the residue around $u = 0$,

$$
\begin{aligned}
\mathcal{B} &= -\oint_{u=0} \frac{\mathrm{d}u}{u} \frac{z_I}{1 - z_I u} \left(\langle 1^I|r|2^J] + [1^I|r|2^J\rangle\right) \\
&= -z_I\left(\langle 1^I|r|2^J] + [1^I|r|2^J\rangle\right).
\end{aligned}
\tag{79}
$$

We see that the little group non-covariant boundary term cancels precisely with the little group non-covariant part of the residue at $z = z_I$. The final amplitude,

$$A_4\left[\mathcal{W}_1^I, \bar{\mathcal{W}}_2^J, \Phi_3, \bar{\Phi}_4\right] = -\left(\frac{\langle 1^I|p_4|2^J] + [1^I|p_4|2^J\rangle + m_4[1^I 2^J] + m_4\langle 1^I 2^J\rangle}{s_{12}s_{14}}\right)\delta^{(2)}(Q^\dagger)\delta^{(2)}(Q)\,, \quad (80)$$

which is in full agreement with (70) calculated by projection.

## 5.3 Amplitude with four massive hypermultiplets

In this section we want to compute four-point hypermultiplet amplitude $A_4\left[\Phi_1, \bar{\Phi}_2, \Phi_3, \bar{\Phi}_4\right]$ in $\mathcal{N} = 2^*$ theory. Similar to the previous section, we will take legs 1 and 3 to be BPS and the rest of them to be anti-BPS, and all external states to be outgoing. Consider BCFW shifts in legs 1 and 3 in terms of the complex parameter $z$ as,

$$\begin{aligned}\hat{p}_1 &= p_1 + zr\,,\\ \hat{p}_3 &= p_3 - zr\,,\end{aligned} \quad (81)$$

the conditions on the null momentum $r$ remain the same.

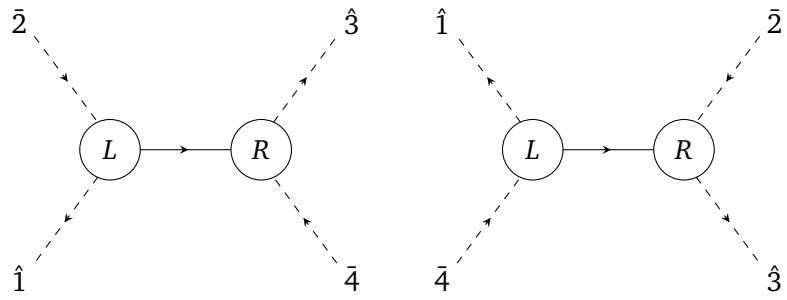

Here both the $s_{12}$-channel and the $s_{14}$-channel will contribute with massive $\mathcal{W}$-boson exchange such that,

$$\begin{aligned}A_4\left[\Phi_1, \bar{\Phi}_2, \Phi_3, \bar{\Phi}_4\right] &= \oint_{z\neq 0}\frac{\mathrm{d}z}{z}\frac{z_{I,1}}{z - z_{I,1}}\int \mathrm{d}^2\eta_{\hat{P}} A_L\left[\bar{\Phi}_4, \hat{\Phi}_1, \mathcal{W}_{\hat{P}}^M\right]\frac{-\epsilon_{MN}}{s_{14}}A_R\left[\bar{\mathcal{W}}_{-\hat{P}}^N, \bar{\Phi}_2, \hat{\Phi}_3\right]\\ &\quad + \oint_{z\neq 0}\frac{\mathrm{d}z}{z}\frac{z_{I,2}}{z - z_{I,2}}\int \mathrm{d}^2\eta_{\hat{P}} A_L\left[\hat{\Phi}_1, \bar{\Phi}_2, \mathcal{W}_{\hat{P}}^M\right]\frac{-\epsilon_{MN}}{s_{12}}A_R\left[\bar{\mathcal{W}}_{-\hat{P}}^N, \hat{\Phi}_3, \bar{\Phi}_4,\right]\,, \quad (82)\end{aligned}$$

with the pole contributions at, $z_{I,1} = \frac{s_{14}}{2r\cdot p_4}$ and $z_{I,2} = \frac{s_{12}}{2r\cdot p_2}$. Let us study factorisation in $s_{14}$-channel first. After contour deformation away from $z = 0$ we can write,

$$A_4^{s_{14}}\left[\Phi_1, \bar{\Phi}_2, \Phi_3, \bar{\Phi}_4\right] = \oint_{z\neq 0}\frac{\mathrm{d}z}{z}\frac{z_{I,1}}{z - z_{I,1}}\int \mathrm{d}^2\eta_{\hat{P}} A_L\left[\bar{\Phi}_4, \hat{\Phi}_1, \mathcal{W}_{\hat{P}}^M\right]\frac{-\epsilon_{MN}}{s_{14}}A_R\left[\bar{\mathcal{W}}_{-\hat{P}}^N, \bar{\Phi}_2, \hat{\Phi}_3\right]\,, \quad (83)$$

where the generalized Mandelstam variable $s_{14} = -\left(p_1 + p_4\right)^2 - \left(m_1 - m_4\right)^2$. The left and right amplitudes can be expressed as,

$$\begin{aligned}A_L\left[\bar{\Phi}_4, \hat{\Phi}_1, \mathcal{W}_{\hat{P}}^M\right] &= \frac{\langle u^{(L)}q\rangle\langle u^{(L)}\hat{P}^M\rangle}{m_1 m_P\langle q|\hat{P}\hat{p}_1|q\rangle}\delta^{(2)}\left(\hat{Q}_L^\dagger\right)\delta\left(\langle q|\hat{p}_1|\hat{Q}_L]\right)\,,\\ A_R\left[\bar{\mathcal{W}}_{-\hat{P}}^N, \bar{\Phi}_2, \hat{\Phi}_3\right] &= -\frac{\langle u^{(R)}q\rangle\langle u^{(R)}\hat{P}^N\rangle}{m_3 m_P\langle q|\hat{P}\hat{p}_3|q\rangle}\delta^{(2)}\left(\hat{Q}_R^\dagger\right)\delta\left(\langle q|\hat{p}_3|\hat{Q}_R]\right)\,. \quad (84)\end{aligned}$$

Similar delta function manipulation and $\eta_{\hat{P}}$ integration gives,

$$
\begin{aligned}
&A_4^{s_{14}}\left[\Phi_1, \bar{\Phi}_2, \Phi_3, \bar{\Phi}_4\right] \\
&= \frac{1}{s_{14}} \oint_{\substack{z \neq 0}} \frac{dz}{z} \frac{z_{I,1}}{z - z_{I,1}} \frac{\langle u^{(R)}q\rangle\langle u^{(L)}q\rangle\langle u^{(L)}\hat{P}K\rangle\langle u^{(R)}\hat{P}_K\rangle}{m_P^2 \langle q|\hat{P}\hat{p}_1|q\rangle\langle q|\hat{P}\hat{p}_3|q\rangle} \frac{\langle u^{(R)}q\rangle\langle u^{(L)}q\rangle}{\left(u_{\hat{P}M}^{(R)} u_{\hat{P}}^{(L)M}\right)} \delta^{(2)}\left(Q^\dagger\right)\delta^{(2)}(Q) \\
&= \frac{1}{s_{14}} \delta^{(2)}\left(Q^\dagger\right)\delta^{(2)}(Q) .
\end{aligned}
$$
(85)

In the penultimate step we have used the spin sum (101) to write, $\langle u^{(L)}\hat{P}^K\rangle\langle u^{(R)}\hat{P}_K\rangle = m_P^2\left(u_{\hat{P}M}^{(R)} u_{\hat{P}}^{(L)M}\right)$ and, $\langle q|\hat{P}\hat{p}_1|q\rangle = \langle u^{(L)}q\rangle^2$, also, $\langle q|\hat{P}\hat{p}_3|q\rangle = \langle u^{(R)}q\rangle^2$ as discussed earlier. It is to be noted that unlike the analysis of amplitudes involving vector multiplets where residue over one channel always contains a pole in another channel, in this case, this does not occur. Therefore we have to consider contributions from both the diagrams given above in the BCFW analysis.

Similarly, from the $s_{12}$-channel computation we get,

$$
A_4^{s_{12}}\left[\Phi_1, \bar{\Phi}_2, \Phi_3, \bar{\Phi}_4\right] = \frac{1}{s_{12}} \delta^{(2)}\left(Q^\dagger\right)\delta^{(2)}(Q) ,
$$
(86)

and the total amplitude,

$$
\begin{aligned}
A_4\left[\Phi_1, \bar{\Phi}_2, \Phi_3, \bar{\Phi}_4\right] &= A_4^{s_{14}}\left[\Phi_1, \bar{\Phi}_2, \Phi_3, \bar{\Phi}_4\right] + A_4^{s_{12}}\left[\Phi_1, \bar{\Phi}_2, \Phi_3, \bar{\Phi}_4\right] \\
&= -\frac{s_{13}}{s_{12}s_{14}} \delta^{(2)}\left(Q^\dagger\right)\delta^{(2)}(Q) .
\end{aligned}
$$
(87)

From the definition of generalized Mandelstam variables and applying momentum conservation, $p_1 + p_2 + p_3 + p_4 = 0$ and central charge conservation $m_1 + m_3 = m_2 + m_4$ we get, $s_{12} + s_{14} = -s_{13}$.

## 5.4 Amplitude with four massive vector multiplets

To calculate the four-point massive SYM amplitude $A_4\left[\mathcal{W}_1^I, \bar{\mathcal{W}}_2^J, \mathcal{W}_3^K, \bar{\mathcal{W}}_4^L\right]$ in $\mathcal{N} = 2^*$ theory, we take deformations in legs labeled by 1 and 2 in terms of the complex parameter $z$,

$$
\begin{aligned}
\hat{p}_1^\mu &= p_1^\mu + z r^\mu, \\
\hat{p}_2^\mu &= p_2^\mu - z r^\mu,
\end{aligned}
$$
(88)

where the complex null vector $r^\mu$ is orthogonal to both the momenta $p_1$ and $p_2$. For this BCFW shift only the $s_{14}$-channel diagram will contribute.

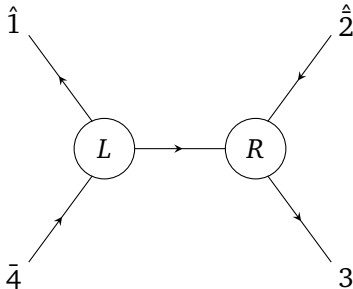

After contour deformation away from $z = 0$, we can write,

$$
A_4\left[\mathcal{W}_1^I, \bar{\mathcal{W}}_2^J, \mathcal{W}_3^K, \bar{\mathcal{W}}_4^L\right] = \oint_{\substack{z \neq 0}} \frac{dz}{z} \frac{z_I}{z - z_I} \int d^2\eta_{\hat{P}} A_L\left[\bar{\mathcal{W}}_4^L, \hat{\mathcal{W}}_1^I, \mathcal{W}_{\hat{P}}^M\right] \frac{-1}{s_{14}} A_R\left[\bar{\mathcal{W}}_{-\hat{P}M}, \hat{\bar{\mathcal{W}}}_2^J, \mathcal{W}_3^K\right],
$$
(89)

where, the generalised Mandelstam variable is $s_{14} = -(p_1 + p_4)^2 - (m_1 - m_4)^2$ and the location of the pole $z_I = \frac{s_{14}}{2r \cdot p_4}$. The left and right amplitudes can be expressed as,

$$
\begin{aligned}
A_L\left[\bar{\mathcal{W}}_4^L, \hat{\mathcal{W}}_1^I, \mathcal{W}_{\hat{P}}^M\right] &= \frac{\mathcal{N}_L}{m_1\langle q|\hat{p}_1 p_4|q\rangle}\delta^{(2)}\left(\hat{Q}_L^\dagger\right)\delta\left(\langle q|\hat{p}_1|\hat{Q}_L]\right), \\
A_R\left[\bar{\mathcal{W}}_{-\hat{P}M}, \hat{\bar{\mathcal{W}}}_2^J, \mathcal{W}_3^K\right] &= \frac{\mathcal{N}_R}{m_2\langle q|\hat{p}_2 p_3|q\rangle}\delta^{(2)}\left(\hat{Q}_R^\dagger\right)\delta\left(\langle q|\hat{p}_2|\hat{Q}_R]\right),
\end{aligned}
\tag{90}
$$

where the full expressions of the numerators in terms of the $u$-spinors,

$$
\begin{aligned}
\mathcal{N}_L &= -\frac{\langle u^{(L)}q\rangle}{m_P}\left(\frac{\langle u^{(L)}4^L\rangle\langle\hat{1}^I\hat{P}^M\rangle}{m_4} + \frac{\langle u^{(L)}\hat{1}^I\rangle\langle 4^L\hat{P}^M\rangle}{m_1}\right) = \frac{\langle u^{(L)}q\rangle}{m_P}\left(u_4^{(L)L}\langle\hat{1}^I\hat{P}^M\rangle + u_{\hat{1}}^{(L)I}\langle 4^L\hat{P}^M\rangle\right), \\
\mathcal{N}_R &= -\frac{\langle u^{(R)}q\rangle}{m_P}\left(\frac{\langle u^{(R)}\hat{2}^J\rangle\langle 3^K\hat{P}_M\rangle}{m_2} + \frac{\langle u^{(R)}3^K\rangle\langle\hat{2}^J\hat{P}_M\rangle}{m_3}\right) = \frac{\langle u^{(R)}q\rangle}{m_P}\left(u_{\hat{2}}^{(R)J}\langle 3^K\hat{P}_M\rangle + u_3^{(R)K}\langle\hat{2}^J\hat{P}_M\rangle\right).
\end{aligned}
\tag{91}
$$

After combining the delta functions and integrating over the $\eta_{\hat{P}}$ variables we get,

$$
\begin{aligned}
&A_4\left[\mathcal{W}_1^I, \bar{\mathcal{W}}_2^J, \mathcal{W}_3^K, \bar{\mathcal{W}}_4^L\right] \\
&= \oint_{z\neq 0}\frac{dz}{z}\frac{z_I}{z - z_I}\Big[\left(u_4^{(L)L}\langle\hat{1}^I\hat{P}^M\rangle + u_{\hat{1}}^{(L)I}\langle 4^L\hat{P}^M\rangle\right)\left(u_{\hat{2}}^{(R)J}\langle 3^K\hat{P}_M\rangle + u_3^{(R)K}\langle\hat{2}^J\hat{P}_M\rangle\right)\left(u_{\hat{P}N}^{(L)}u_{\hat{P}}^{(R)N}\right)\Big] \\
&\qquad\qquad \times \frac{\delta^{(2)}\left(Q^\dagger\right)\delta^{(2)}(Q)}{s_{12}s_{14}m_P^2}.
\end{aligned}
\tag{92}
$$

Using multiple Schouten identities, and after nice cancellations between various terms, finally, we have,

$$
\begin{aligned}
A_4\left[\mathcal{W}_1^I, \bar{\mathcal{W}}_2^J, \mathcal{W}_3^K, \bar{\mathcal{W}}_4^L\right] = -\oint_{z\neq 0}\frac{dz}{z}\frac{z_I}{z - z_I}\Big(&\langle\hat{1}^I\hat{2}^J\rangle[3^K 4^L] + \langle\hat{1}^I 3^K\rangle[\hat{2}^J 4^L] + \langle\hat{1}^I 4^L\rangle[\hat{2}^J 3^K] \\
&+ \langle\hat{2}^J 3^K\rangle[\hat{1}^I 4^L] + \langle\hat{2}^J 4^L\rangle[\hat{1}^I 3^K] + \langle 3^K 4^L\rangle[\hat{1}^I\hat{2}^J]\Big)\frac{\delta^{(2)}(Q^\dagger)\delta^{(2)}(Q)}{s_{12}s_{41}}.
\end{aligned}
\tag{93}
$$

Here we see again that before performing the $z$ integral, the expression inside the integral is the shifted version of the answer we obtained from projection. This suggests that the little group non-covariant part of the residue at $z = z_I$ cancels precisely with the little group non-covariant contribution from the pole at infinity. Thus little group non-covariance is useful to deduce the amplitude here before performing the explicit $z$ integration.

## 6 Conclusions

Massless spinor helicity formalism has been playing a central role in the computation of on-shell amplitudes in massless field theories. The extension of this formalism for massive field theories is an essential and obvious step. There has been some progress in this direction already. We have explored the application of the massive spinor helicity formalism to four dimensional $\mathcal{N} = 2^*$ theory at an arbitrary point in the Coulomb branch moduli space. We used two different techniques to compute three and four point amplitudes in this theory. The first method is to write the amplitude in terms of the $u$-spinor which is amenable to setting up the BCFW formulation. Another method is to use the projection method to compute $\mathcal{N} = 2^*$ amplitudes from $\mathcal{N} = 4$ SYM amplitudes. The four point amplitudes involving four massive

hypermultiplets (68), four massive vector multiplets (69), and two massive vector multiplets and two massive hypermultiplets (70) are computed using the projection method. We then proceeded with the BCFW shift for four point amplitudes. One would have naively thought that like the amplitudes, the BCFW analysis could also follow trivially by projection. However, that is not the case because in the $\mathcal{N} = 4$ theory the usual BCFW shift involves the Grassmann variable $\eta_I^2$. However, in the projection method, this variable is projected out. As a result the BCFW rules for $\mathcal{N} = 2^*$ do not descend down in a trivial way from $\mathcal{N} = 4$ theory. In fact, this difference also shows up in the integrand, which has a pole at infinity in the case of $\mathcal{N} = 2^*$. Our BCFW shift is not little group covariant but of course, the final amplitudes are. While this is expected, the contribution of the pole at infinity is instrumental in restoring the little group covariance of the amplitude. We believe this feature may be generic for theories for which BCFW shifts are not little group covariant. It appears that the non-covariant shifts generate good tests for the covariance of the integrated amplitudes. For example, in (78), the integrand of the contour integral already has the correct little group covariant form of the amplitude, if we replace shifted variables with unshifted ones. We believe that this feature may extend beyond the $\mathcal{N} = 2^*$ theory and may suggest the utility of the little group non-covariant shifts. Also, as the recursive structure of amplitudes is nontrivial due to pole at infinity, it will be interesting to study higher point amplitudes with these massive BCFW shifts.

It would be interesting to extend this analysis to loop amplitudes. Additionally, it would be worthwhile to derive these results from higher dimensional spinor helicity formalism [4, 11] so that a unified structure could be uncovered in the computations of scattering amplitudes of massless and massive fields. It would be also interesting to see if the amplitudes of $\mathcal{N} = 2^*$ theory can be put in a CHY-like [9] formulation.

## Acknowledgements

APS thanks Sourav Ballav and Arnab Rudra for useful discussions. DPJ acknowledges support from SERB grant CRG/2018/002835.

## A  Notations and conventions

### A.1  Spinor helicity variables for massive particles

We use the "mostly positive" signature for the metric $\eta_{\mu\nu} = \text{diag}(-1, +1, +1, +1)$ in 4-space-time dimension and the momentum bi-spinor for the massive particle for mass $m$ is given by,

$$p_{\alpha\dot{\beta}} = p^\mu \sigma_{\mu,\alpha\dot{\beta}} = -\sum_I |p^I]_\alpha \langle p_I|_{\dot{\beta}} = \sum_I |p_I]_\alpha \langle p^I|_{\dot{\beta}} \,, \tag{94}$$

we can also write,

$$p^{\dot{\alpha}\beta} = p^\mu \bar{\sigma}_\mu^{\dot{\alpha}\beta} = -\sum_I |p_I\rangle^{\dot{\alpha}} [p^I|^\beta = \sum_I |p^I\rangle^{\dot{\alpha}} [p_I|^\beta \,. \tag{95}$$

Here, $I = \{1, 2\}$ is the $SU(2)$ little group index for the massive particle and $\{\alpha, \dot{\beta}\}$ are the usual $SL(2, \mathbb{C})$ spinor indices. The $SU(2)$ little group indices can be raised and lowered through,

$$\epsilon^{IJ} = -\epsilon_{IJ} = \begin{pmatrix} 0 & 1 \\ -1 & 0 \end{pmatrix}, \tag{96}$$

as follows,

$$|p^I]_\alpha = \epsilon^{IJ} |p_J]_\alpha \qquad \langle p_I|_{\dot{\beta}} = \epsilon_{IJ} \langle p^J|_{\dot{\beta}} \,. \tag{97}$$

The determinant of the momentum bi-spinor gives,

$$\det p = -p^\mu p_\mu = m^2 \,. \tag{98}$$

The bi-linear product of the spinors is given as,

$$\langle p^I p^J \rangle = m\epsilon^{IJ} \,, \quad [p^I p^J] = -m\epsilon^{IJ} \,. \tag{99}$$

These spinors also satisfy the Weyl equation,

$$
\begin{aligned}
p|p^I] &= -m|p^I\rangle \,, & p|p^I\rangle &= -m|p^I] \,, \\
[p^I|p &= m\langle p^I| \,, & \langle p^I|p &= m[p^I| \,,
\end{aligned}
\tag{100}
$$

and spin sums,

$$
\begin{aligned}
|p_I]_\alpha [p^I|^\beta &= -|p^I]_\alpha [p_I|^\beta = m\delta_\alpha^\beta \,, \\
|p_I\rangle^{\dot\alpha} \langle p^I|_{\dot\beta} &= -|p^I\rangle^{\dot\alpha} \langle p_I|_{\dot\beta} = -m\delta_{\dot\beta}^{\dot\alpha} \,.
\end{aligned}
\tag{101}
$$

Some useful identities are listed below,

$$
\begin{aligned}
2p.q &= \langle p_I q_J \rangle [p^I q^J] \,, & 2m_p m_q &= \langle p_I q_J \rangle \langle p^I q^J \rangle = [p_I q_J][p^I q^J] \,, \\
\langle q^I|pp|k^J \rangle &= -p^2 \langle q^I k^J \rangle \,, & [q^I|pp|k^J] &= -p^2 [q^I k^J] \,,
\end{aligned}
\tag{102}
$$

$$\langle q^I|p|k^J] = [k^J|p|q^I\rangle \,. \tag{103}$$

The high energy limit of the massive spinors and the Grassmann variables give,

$$|p^+] \to |p] \,, \quad |p^-] \to 0 \,, \quad |p^+\rangle \to 0 \,, \quad |p^-\rangle \to -|p\rangle \,, \quad \eta_- \to \eta \,, \quad \eta_+ \to \hat\eta \,. \tag{104}$$

We have used the following analytic continuation for the massive spinors and the corresponding variable,

$$|-P^I] = i|P^I] \,, \quad |-P^I\rangle = i|P^I\rangle \,, \quad \eta_{-P}^I = i\eta_P^I \,, \tag{105}$$

similarly in the massless case,

$$|-p] = i|p] \,, \quad |-p\rangle = i|p\rangle \,, \quad \eta_{-p} = i\eta_p \,, \quad \eta_{-p}^\dagger = i\eta_p^\dagger \,. \tag{106}$$

The generalized Mandelstam variables are defined as,

$$s_{ij} = -(p_i + p_j)^2 - (m_i \pm m_j)^2 \,, \tag{107}$$

where the masses are added if the states are both BPS/anti-BPS and subtracted if they are different.

The BPS condition reads,

$$P_i Q_i^{\dagger a} = \pm m_i Q_{ia+2} \,, \tag{108}$$

for $\mathcal{N} = 4$ supersymmetry and $a+2 \to a+1$ on the right hand side for $\mathcal{N} = 2$ supersymmetry. Plus sign here holds for BPS legs whereas the minus sign holds for anti-BPS legs. The supercharges $Q_i^{\dagger a}$ and $Q_{ia+2}$ are defined for each leg and throughout the paper we have considered total supercharges. Our convention for the total supercharges follows that of [18] where,

$$
\begin{aligned}
\frac{1}{\sqrt{2}} Q^{\dagger a} &= -\sum_i \eta_{iI}^a |i^I\rangle - \sum_j \eta_{jI}^a |j^I\rangle + \sum_k \eta_k^a |k\rangle \,, \\
\frac{1}{\sqrt{2}} Q_{a+2} &= \sum_i \eta_{iI} |i^I] - \sum_j \eta_{jI} |j^I] + \sum_k \tilde\eta^{\dagger a} \,,
\end{aligned}
\tag{109}
$$

where $i$ runs over all the BPS legs, $j$ runs over all the anti-BPS legs and $k$ runs over the massless legs and $a+2 \to a+1$ on the left hand side for $\mathcal{N} = 2$ supersymmetry

# B Useful calculations

## B.1 Three point amplitudes in terms of $u$-spinors

In this section we illustrate how to express the massive three-point amplitudes of the $\mathcal{N} = 2^*$ theory in a simpler form using the $u$-spinors. We can use (99) to write ,

$$m_3 = -\frac{1}{2}\epsilon_{JK}\langle 3^J 3^K\rangle. \tag{110}$$

Let us start with the expression of the amplitude, $A_3\left[\mathcal{W}_1^I, \bar{\Phi}_2, \Phi_3\right]$, for which the numerator can be simplified by using the Schouten identity and the $u$-spinors,

$$\begin{aligned}
\langle q|p_3|1^I] + m_3\langle q1^I\rangle &= -\langle q3_J\rangle[3^J 1^I] + \frac{1}{2}\epsilon_{JK}(\langle q3^J\rangle\langle 3^K 1^I\rangle + \langle q3^K\rangle\langle 1^I 3^J\rangle) \\
&= \frac{\langle qu\rangle\langle 1^I u\rangle}{m_1}. \tag{111}
\end{aligned}$$

Following the three particle special kinematics, we have used the identity, $[3^J 1^I] + \langle 3^J 1^I\rangle = u_3^J u_1^I$ to get,

$$A_3\left[\mathcal{W}_1^I, \bar{\Phi}_2, \Phi_3\right] = -\left(\frac{\langle uq\rangle\langle u1^I\rangle}{m_1\langle q|p_1 p_3|q\rangle}\right)\delta^{(2)}(Q)\delta(\langle qQ^\dagger\rangle), \tag{112}$$

Now to represent the amplitude, $A_3\left[\mathcal{W}_1^I, \bar{\mathcal{W}}_2^J, \mathcal{W}_3^K\right]$, in terms of the $u$-spinor, we need to do a few manipulations. By multiplying (111) with $\frac{\langle 2^J 3^K\rangle}{m_3}$ and using the Schouten identity we can write,

$$\frac{\langle q|p_3|1^I]\langle 2^J 3^K\rangle}{m_3} + \frac{\langle q|p_3|3^K]\langle 1^I 2^J\rangle}{m_3} = \frac{\langle uq\rangle\langle u1^I\rangle\langle 2^J 3^K\rangle}{m_1 m_3} - \langle q2^J\rangle\langle 1^I 3^K\rangle. \tag{113}$$

We can then express the term $\langle q|p_3|2^J] - m_3\langle q2^J\rangle$, as follows,

$$\frac{\langle q|p_3|2^J]\langle 1^I 3^K\rangle}{m_3} = \frac{\langle uq\rangle\langle u2^J\rangle\langle 1^I 3^K\rangle}{m_2 m_3} + \langle q2^J\rangle\langle 1^I 3^K\rangle. \tag{114}$$

Now clubbing the above terms together we have,

$$A_3\left[\mathcal{W}_1^I, \bar{\mathcal{W}}_2^J, \mathcal{W}_3^K\right] = -\left(\frac{\langle uq\rangle\langle u1^I\rangle\langle 2^J 3^K\rangle}{m_1 m_3} + \frac{\langle uq\rangle\langle u2^J\rangle\langle 1^I 3^K\rangle}{m_2 m_3}\right)\frac{\delta^{(2)}(Q)\delta(\langle qQ^\dagger\rangle)}{\langle q|p_1 p_3|q\rangle}. \tag{115}$$

We note that while in the original form of the three point amplitudes the BCFW manipulations are not obvious, these representations of three point amplitudes in terms of $u$-spinor simplify the manipulations significantly.

## B.2 Some detailed BCFW calculations

In this section, we will explain some calculations used in section 5. For example, using the $u$-spinor as shown in (37), we can write,

$$\hat{p}_1 = \frac{1}{|u_1|}(|\hat{1}^w\rangle[u^{(L)}| + |u^{(L)}\rangle[\hat{1}^w|), \tag{116}$$

$$p_4 = \frac{1}{|u_4|}(|4^w\rangle[u^{(L)}| - |u^{(L)}\rangle[4^w|), \tag{117}$$

where $|i^w\rangle = \hat{w}_{iI}|i^I\rangle$. With the above representations of the momenta corresponding to the left amplitude and using the Schouten identity, we have,

$$
\begin{aligned}
\langle q|\hat{p}_1 p_4|q\rangle &= \frac{\langle qu^{(L)}\rangle \langle qu^{(L)}\rangle}{|u_1||u_4|}(\langle\hat{1}^w 4^w\rangle + [\hat{1}^w 4^w]) \\
&= \langle u^{(L)}q\rangle^2 .
\end{aligned}
\tag{118}
$$

Similarly, one can show, $\langle q|\hat{p}_2 p_3|q\rangle = \langle u^{(R)}q\rangle^2$.

Another crucial calculation in (77) of combining the delta functions involves,

$$
\delta\left(u_{4,K}\langle 4^K|\hat{p}_1|\hat{Q}_R]\right) = \left(\frac{\alpha}{\beta\gamma}\right)\delta\left(u_{4,K}\langle 4^K|\hat{p}_1|Q]\right) .
\tag{119}
$$

The definitions of $\alpha$ and $\beta$ come from the Schouten identity,

$$
\langle u^{(L)}q\rangle\langle u^{(R)}| + \langle qu^{(R)}\rangle\langle u^{(L)}| + \langle u^{(R)}u^{(L)}\rangle\langle q| = 0 ,
\tag{120}
$$

such that the relation, $u^{(R)}_{\hat{P}M} = \alpha u^{(R)}_{\hat{P}M} + \beta q$ holds with,

$$
\alpha = \frac{u^{(R)}_{3,K}\langle 3^K q\rangle}{u^{(L)}_{4,M}\langle 4^M q\rangle} , \qquad \beta = \frac{u^{(R)}_{3,K}\langle 3^K 4^M\rangle u^{(L)}_{4,M}}{\langle q4^M\rangle u^{(L)}_{4,M}} .
\tag{121}
$$

In a similar procedure, starting with the expression $\langle q|\hat{p}_1|u^{(L)}]\langle q|\hat{P}$, and using the Schouten identity, we can write,

$$
\langle q|\frac{\hat{P}}{m_P} = \gamma u^{(L)}_{4,M}\langle 4^M|\frac{\hat{p}_1}{m_1} + \lambda\langle q|\frac{\hat{p}_1}{m_1} ,
\tag{122}
$$

where, the coefficient $\lambda$ is not relevant for our purpose, and

$$
\gamma = \frac{\langle q|\hat{p}_1 p_4|q\rangle}{m_1 m_P u^{(L)}_{4,M}\langle 4^M q\rangle} .
\tag{123}
$$

With these definitions of $\alpha$, $\beta$, and $\gamma$ one can obtain (77).

## C  $\mathcal{N} = 4$

BCFW recursion relations for $\mathcal{N} = 4$ SYM amplitudes in chiral superspace are very well developed [7]. Here we present some BCFW analysis for four-point amplitudes in $\mathcal{N} = 4$ SYM theory in non-chiral superspace.

### C.1  Massless amplitude

We choose deformations in external states 1 and 2 which are given by,

$$
\hat{p}_1 = p_1 + zr , \qquad \hat{p}_2 = p_2 - zr ,
\tag{124}
$$

with the conditions that $p_1 \cdot r = p_2 \cdot r = r^2 = 0$.

Motivated by the conservation of momenta and supercharges, we consider the following shifts in the spinor and Grassmanian variables,

$$
\begin{aligned}
|\hat{1}] &= |1] + z|2] , \\
|\hat{2}\rangle &= |2\rangle - z|1\rangle , \\
\hat{\eta}_1^a &= \eta_1^a + z\eta_2^a , \\
\hat{\tilde{\eta}}_2^{\dagger a} &= \tilde{\eta}_2^{\dagger a} - z\tilde{\eta}_1^{\dagger a} , \qquad a = 1, 2 .
\end{aligned}
\tag{125}
$$

It can be checked that under the above shifts supersymmetric charges, $\mathcal{Q}^{\dagger} = \prod_{a=1,2} \sum_i |i\rangle \eta_i^a$, and $\mathcal{Q} = \prod_{a=1,2} \sum_i |i] \hat{\eta}_i^{\dagger\,a}$ remain invariant.

Let us consider the u-channel factorization.

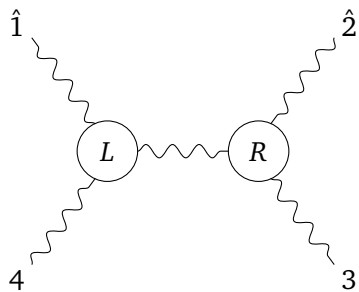

In this case the four-point amplitude can be obtained by,

$$
\begin{aligned}
\mathcal{A}_4\big[G_1, G_2, G_3, G_4\big] &= \oint_{\{z=0\}} \frac{dz}{z} \mathcal{A}(z) \\
&= \oint_{\{z=0\}} \frac{dz}{z} \int d^2\eta_{\hat{P}} \mathcal{A}_L(z) \frac{1}{\hat{s}_{14}} \mathcal{A}_R(z) \\
&= -\oint_{\{z=0\}} \frac{dz}{z} \frac{z_I}{z - z_I} \int d^2\eta_{\hat{P}} \mathcal{A}_L(z) \frac{1}{s_{14}} \mathcal{A}_R(z) \\
&= \int d^2\eta_{\hat{P}} \mathcal{A}_L(z_I) \frac{1}{s_{14}} \mathcal{A}_R(z_I),
\end{aligned} \tag{126}
$$

where $z_I = \frac{s_{14}}{2r \cdot p_4}$. Here we have assumed there is no pole at infinity and this assumption is justified from the large z behavior of the amplitude.

Three-point sub-amplitudes are given by,

$$
\mathcal{A}_L^{\mathrm{MHV}}\big[G_4, \hat{G}_1, G_{\hat{P}}\big] = \frac{1}{\langle 41\rangle \langle 1\hat{P}\rangle \langle \hat{P}4\rangle} \delta^{(4)}\big(\hat{\mathcal{Q}}_L^{\dagger}\big) \delta^{(2)}\big(\langle 41\rangle \tilde{\eta}_{\hat{P}}^{\dagger a} + \langle 1\hat{P}\rangle \tilde{\eta}_4^{\dagger a} + \langle \hat{P}4\rangle \tilde{\eta}_1^{\dagger a}\big),
$$
$$
\mathcal{A}_R^{\mathrm{anti\text{-}MHV}}\big[G_{-\hat{P}}, \hat{G}_2, G_3\big] = \frac{1}{[\hat{P}2][23][3\hat{P}]} \delta^{(4)}\big(\hat{\mathcal{Q}}_R\big) \delta^{(2)}\big([\hat{P}2]\eta_3^a + [23]\eta_{\hat{P}}^a + [3\hat{P}]\eta_2^a\big). \tag{127}
$$

First we perform the $\eta_{\hat{P}}$ integration. Solutions to $\eta_{\hat{P}}$ and $\tilde{\eta}_{\hat{P}}^{\dagger}$ are available from the delta functions. On the support of $\delta^{(2)}\big([\hat{P}2]\eta_3^a + [23]\eta_{\hat{P}}^a + [3\hat{P}]\eta_2^a\big)$ we get,

$$
\begin{aligned}
\delta^{(4)}\big(\hat{\mathcal{Q}}_L^{\dagger}\big) &= \delta^{(4)}\left(\prod_{a=1,2}\big(|4\rangle \eta_4^a + |1\rangle \hat{\eta}_1^a + |\hat{P}\rangle \eta_{\hat{P}}^a\big)\right) \\
&= \delta^{(4)}\left(\prod_{a=1,2}\big(|4\rangle \eta_4^a + |1\rangle \hat{\eta}_1^a - |\hat{P}\rangle \frac{1}{[23]}\big([\hat{P}2]\eta_3^a + [3\hat{P}]\eta_2^a\big)\big)\right) \\
&= \delta^{(4)}\left(\prod_{a=1,2} \sum_{i=1}^4 |i\rangle \eta_i^a\right). \tag{128}
\end{aligned}
$$

To go from second equality to the last one we have used momentum conservation. Similar

manipulations holds for $\delta^{(4)}\big(\hat{\mathcal{Q}}_R\big)$ on the support of the other delta function,

$$
\int d^2\eta_{\hat{P}} d^2\tilde{\eta}_{\hat{P}}^{\dagger} \delta^{(4)}\big(\hat{\mathcal{Q}}_L^{\dagger}\big)\delta^{(2)}\big(\langle 41\rangle\tilde{\eta}_{\hat{P}}^{\dagger a}+\langle 1\hat{P}\rangle\tilde{\eta}_4^{\dagger a}+\langle\hat{P}4\rangle\tilde{\eta}_1^{\dagger a}\big) \tag{129}
$$

$$
\times\,\delta^{(4)}\big(\hat{\mathcal{Q}}_R\big)\delta^{(2)}\big([\hat{P}2]\eta_3^a+[23]\eta_{\hat{P}}^a+[3\hat{P}]\eta_2^a\big) \;=\; \langle 41\rangle^2[23]^2\delta^{(4)}\big(\mathcal{Q}^{\dagger}\big)\delta^{(4)}(\mathcal{Q})\,.
$$

Therefore,

$$
\begin{aligned}
\int d^2\eta_{\hat{P}}\mathcal{A}_L(z_I)\mathcal{A}_R(z_I) &= \frac{\langle 41\rangle[23]}{\langle 1\hat{P}\rangle\langle\hat{P}4\rangle[\hat{P}2][3\hat{P}]}\delta^{(4)}\big(\mathcal{Q}^{\dagger}\big)\delta^{(4)}(\mathcal{Q}) \\
&= \frac{1}{s_{12}}\delta^{(2)}\big(\mathcal{Q}^{\dagger}\big)\delta^{(2)}(\mathcal{Q})\,.
\end{aligned} \tag{130}
$$

The four point amplitude is then,

$$
\mathcal{A}_4\big[G_1,G_2,G_3,G_4\big]=\frac{1}{s_{12}s_{14}}\delta^{(4)}\big(\mathcal{Q}^{\dagger}\big)\delta^{(4)}(\mathcal{Q})\,. \tag{131}
$$

## C.2 Massive amplitude with massless exchange

Here we consider the external states to be massive $\mathcal{W}$-bosons and in the intermediate channel massless vector multiplet is exchanged. We want to find out the four-point amplitude $\mathcal{A}_4\big(\mathcal{W}_1,\bar{\mathcal{W}}_2,\mathcal{W}_3,\bar{\mathcal{W}}_4\big)$. We consider shifts in legs 1 and 2 as before.

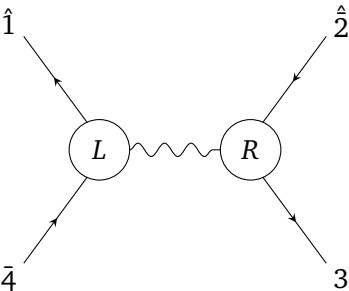

The left and right three-point amplitudes are,

$$
\begin{aligned}
\mathcal{A}_L\big(\bar{\mathcal{W}}_4,\hat{\mathcal{W}}_1,G_{\hat{P}}\big) &= \frac{-\hat{x}_{14}}{m_1^3\langle q\hat{P}\rangle^2}\delta^{(4)}\big(\hat{\mathcal{Q}}_L^{\dagger}\big)\delta^{(2)}\big(\langle q|\hat{p}_1|\hat{\mathcal{Q}}_L]\big) = \frac{-\hat{x}_{14}}{m_1\langle q\hat{P}\rangle^2}\delta^{(4)}\big(\hat{\mathcal{Q}}_L\big)\delta^{(2)}\big(\langle q\hat{\mathcal{Q}}_L^{\dagger}\rangle\big)\,, \\
\mathcal{A}_R\big(G_{-\hat{P}},\hat{\bar{\mathcal{W}}}_2,\mathcal{W}_3\big) &= \frac{-\hat{x}_{23}}{m_2^3\langle q\hat{P}\rangle^2}\delta^{(4)}\big(\hat{\mathcal{Q}}_R^{\dagger}\big)\delta^{(2)}\big(\langle q|\hat{p}_2|\hat{\mathcal{Q}}_R]\big) = \frac{-\hat{x}_{23}}{m_3\langle q\hat{P}\rangle^2}\delta^{(4)}\big(\hat{\mathcal{Q}}_R\big)\delta^{(2)}\big(\langle q\hat{\mathcal{Q}}_R^{\dagger}\rangle\big)\,.
\end{aligned} \tag{132}
$$

Central charge conservation in the half-BPS limit implies that,

$$
m_1=m_4\,,\qquad m_2=m_3\,. \tag{133}
$$

Using the above relations we can determine $x$ factors,

$$
\begin{aligned}
\frac{\hat{p}_1}{m_1}|\hat{P}\rangle=\hat{x}_{14}|\hat{P}] &\;\Rightarrow\; \hat{x}_{14}=\frac{m_1\langle q\hat{P}\rangle}{\langle q|\hat{p}_1|\hat{P}]}=\frac{[\rho|\hat{p}_1|\hat{P}\rangle}{m_1[\rho\hat{P}]}\,, \\
\frac{\hat{p}_2}{m_2}|\hat{P}\rangle=\hat{x}_{23}|\hat{P}] &\;\Rightarrow\; \hat{x}_{23}=\frac{m_2\langle q\hat{P}\rangle}{\langle q|\hat{p}_2|\hat{P}]}=\frac{[\rho|\hat{p}_2|\hat{P}\rangle}{m_2[\rho\hat{P}]}\,.
\end{aligned} \tag{134}
$$

Product of the two three-point amplitudes gives,

$$\int d^2\eta_{\hat{P}} \mathcal{A}_L\left(\bar{\mathcal{W}}_4, \hat{\mathcal{W}}_1, G_{\hat{P}}\right) \mathcal{A}_R\left(G_{-\hat{P}}, \hat{\bar{\mathcal{W}}}_2, \mathcal{W}_3\right)$$

$$= \int d^2\eta_{\hat{P}} \frac{\hat{x}_{14}\hat{x}_{23}}{m_1^3 m_2^3 \langle q\hat{P}\rangle^4} \delta^{(4)}\left(\hat{\mathcal{Q}}_L^\dagger\right) \delta^{(2)}\left(\langle q|\hat{p}_1|\hat{\mathcal{Q}}_L]\right) \delta^{(4)}\left(\hat{\mathcal{Q}}_R^\dagger\right) \delta^{(2)}\left(\langle q|\hat{p}_2|\hat{\mathcal{Q}}_R]\right)$$

$$= \int d^2\eta_{\hat{P}} \frac{\hat{x}_{14}\hat{x}_{23}}{m_1^3 m_2 \langle q\hat{P}\rangle^4} \delta^{(4)}\left(\mathcal{Q}^\dagger\right) \delta^{(2)}\left(\langle q|\hat{p}_1|\mathcal{Q}]\right) \frac{1}{m_1^2 \langle q\hat{P}\rangle^2}$$

$$\times \delta^{(2)}\left(\langle\hat{P}|\hat{p}_1|\hat{\mathcal{Q}}_R]\right) \delta^{(2)}\left(\langle q|\hat{p}_1|\hat{\mathcal{Q}}_R]\right) \delta^{(2)}\left(\langle q\hat{\mathcal{Q}}_R^\dagger\rangle\right). \tag{135}$$

Now, on the support of $\delta^{(2)}\left(\mathcal{Q}^\dagger\right)$ we have,

$$\langle\hat{P}\hat{\mathcal{Q}}_L^\dagger\rangle + \langle\hat{P}\hat{\mathcal{Q}}_R^\dagger\rangle = 0$$

$$\Rightarrow \quad \frac{1}{m_1}\langle\hat{P}|\hat{p}_1|\hat{\mathcal{Q}}_L] - \frac{1}{m_2}\langle\hat{P}|\hat{p}_2|\hat{\mathcal{Q}}_R] = 0. \tag{136}$$

Using Eq.(134) we get,

$$\langle\hat{P}|\hat{p}_1|\hat{\mathcal{Q}}_R] = \frac{m_1\hat{x}_{14}}{m_2\hat{x}_{23}}\langle\hat{P}|\hat{p}_2|\hat{\mathcal{Q}}_R]. \tag{137}$$

From the above two equations we then obtain,

$$\langle\hat{P}|\hat{p}_1|\mathcal{Q}] = \left(1 + \frac{\hat{x}_{23}}{\hat{x}_{14}}\right)\langle\hat{P}|\hat{p}_1|\hat{\mathcal{Q}}_R]. \tag{138}$$

Therefore,

$$\mathcal{A}_L\left(\bar{\mathcal{W}}_4, \hat{\mathcal{W}}_1, G_{\hat{P}}\right) \mathcal{A}_R\left(G_{-\hat{P}}, \hat{\bar{\mathcal{W}}}_2, \mathcal{W}_3\right) \tag{139}$$

$$= \frac{-\hat{x}_{14}\hat{x}_{23}}{m_1^3 m_2 \langle q\hat{P}\rangle^4}\left(1 + \frac{\hat{x}_{23}}{\hat{x}_{14}}\right)^{-2} \delta^{(4)}\left(\mathcal{Q}^\dagger\right) \delta^{(4)}\left(\mathcal{Q}\right) \int d^2\eta_{\hat{P}} \delta^{(2)}\left(\langle q|\hat{p}_1|\hat{\mathcal{Q}}_R]\right) \delta^{(2)}\left(\langle q\hat{\mathcal{Q}}_R^\dagger\rangle\right).$$

Performing the $\eta_{\hat{P}}$ integral we get,

$$\int d^4\eta_{\hat{P}} \delta^{(2)}\left(\langle q|\hat{p}_1|\hat{\mathcal{Q}}_R]\right) \delta^{(2)}\left(\langle q\hat{\mathcal{Q}}_R^\dagger\rangle\right) = \left(\langle q|\hat{p}_1\hat{P}|q\rangle\right)^2 = m_1^2 \frac{\langle q\hat{P}\rangle^4}{\hat{x}_{14}^2}. \tag{140}$$

Then the integration in the complex plane is given by,

$$\mathcal{A}_4\left(\mathcal{W}_1, \bar{\mathcal{W}}_2, \mathcal{W}_3, \bar{\mathcal{W}}_4\right)$$

$$= -\oint_{\{z=0\}} \frac{dz}{z}\frac{z_I}{z - z_I}\frac{1}{m_1 m_2}\frac{1}{s_{14}}\frac{\hat{x}_{23}}{\hat{x}_{14}}\left(1 + \frac{\hat{x}_{23}}{\hat{x}_{14}}\right)^{-2} \delta^{(4)}\left(\mathcal{Q}^\dagger\right) \delta^{(4)}\left(\mathcal{Q}\right)$$

$$= -\oint_{\{z=0\}} \frac{dz}{z}\frac{z_I}{z - z_I}\frac{1}{m_1 m_2}\frac{1}{s_{14}}\left[\frac{\hat{x}_{14}}{\hat{x}_{23}}\left(1 + \frac{\hat{x}_{23}}{\hat{x}_{14}}\right)^2\right]^{-1} \delta^{(4)}\left(\mathcal{Q}^\dagger\right) \delta^{(4)}\left(\mathcal{Q}\right). \tag{141}$$

The expression inside the box brackets can be manipulated as follows,

$$\frac{\hat{x}_{14}}{\hat{x}_{23}}\left(1 + \frac{\hat{x}_{23}}{\hat{x}_{14}}\right)^2 = \frac{\hat{x}_{14}}{\hat{x}_{23}} + \frac{\hat{x}_{23}}{\hat{x}_{14}} + 2$$

$$= \frac{[\rho|\hat{p}_1|\hat{P}\rangle[\hat{P}|\hat{p}_2|q\rangle}{m_1 m_2[\rho\hat{P}]\langle q\hat{P}\rangle} + \frac{[\rho|\hat{p}_2|\hat{P}\rangle[\hat{P}|\hat{p}_1|q\rangle}{m_1 m_2[\rho\hat{P}]\langle q\hat{P}\rangle} + 2$$

$$= \frac{-2p_1 \cdot p_2}{m_1 m_2} + 2$$

$$= \frac{s_{12}}{m_1 m_2}. \tag{142}$$

To go from the second equality to the third, we have used the fact that $\hat{P} \cdot \hat{p}_1 = 0$ and $\hat{P} \cdot \hat{p}_2 = 0$ on $z = z_I$ which implies the momenta bispinors anticommute. Then the four-point amplitude is given by,

$$\mathcal{A}_4\left(\mathcal{W}_1, \bar{\mathcal{W}}_2, \mathcal{W}_3, \bar{\mathcal{W}}_4\right) = \frac{1}{s_{12} s_{14}} \delta^{(4)}\left(\mathcal{Q}^{\dagger}\right) \delta^{(4)}(\mathcal{Q}). \tag{143}$$

# D  Amplitudes with massless exchange

In this section we show evaluation of some four-point amplitudes with massive external states where the intermediate propagators are massless.

## D.1  2 vector and 2 hypermultiplet amplitude

We consider the four-point amplitude $A_4\left[\mathcal{W}_1^I, \bar{\mathcal{W}}_2^J, \Phi_3, \bar{\Phi}_4\right]$. For simplicity we apply shifts in the hypermultiplet legs 3 and 4, such that,

$$\hat{p}_3 = p_3 - zr, \qquad \hat{p}_4 = p_4 + zr. \tag{144}$$

From the on-shell condition we have $z_I = \frac{s_{14}}{2r \cdot p_1}$.

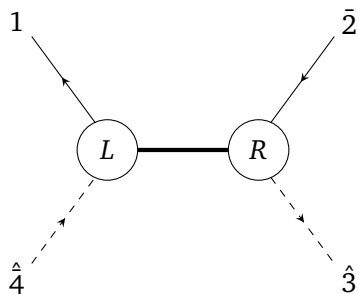

The three-point sub-amplitudes can be expressed as,

$$A_L\left[\hat{\bar{\Phi}}_4, \mathcal{W}_1^I, \Phi_{\hat{P}}\right] = \frac{[\hat{P}1^I]}{\langle q|p_1|\hat{P}]} \delta^{(2)}\left(\hat{Q}_L\right) \delta\left(\langle q\hat{Q}_L^{\dagger}\rangle\right) = \frac{\hat{x}_{14}[\hat{P}1^I]}{m_1\langle q\hat{P}\rangle} \delta^{(2)}\left(\hat{Q}_L\right) \delta\left(\langle q\hat{Q}_L^{\dagger}\rangle\right),$$

$$A_R\left[\Phi_{-\hat{P}}, \bar{\mathcal{W}}_2^J, \hat{\Phi}_3\right] = \frac{[\hat{P}2^J]}{\langle q|p_2|\hat{P}]} \delta^{(2)}\left(\hat{Q}_R\right) \delta\left(\langle q\hat{Q}_R^{\dagger}\rangle\right) = \frac{\hat{x}_{23}[\hat{P}2^J]}{m_2\langle q\hat{P}\rangle} \delta^{(2)}\left(\hat{Q}_R\right) \delta\left(\langle q\hat{Q}_R^{\dagger}\rangle\right), \tag{145}$$

where $\hat{x}_{14}$ and $\hat{x}_{23}$ are given by,

$$\hat{x}_{14} = \frac{[\rho|p_1|\hat{P}\rangle}{m_1[\rho\hat{P}]} = \frac{m_1\langle q\hat{P}\rangle}{\langle q|p_1|\hat{P}]}, \qquad \hat{x}_{23} = \frac{[\rho|p_2|\hat{P}\rangle}{m_2[\rho\hat{P}]} = \frac{m_2\langle q\hat{P}\rangle}{\langle q|p_2|\hat{P}]}. \tag{146}$$

Using BCFW analysis we can write,

$$A_4\left[\mathcal{W}_1^I, \bar{\mathcal{W}}_2^J, \Phi_3, \bar{\Phi}_4\right]$$

$$= -\oint_{\{z=0\}} \frac{dz}{z} \frac{z_I}{z - z_I} A_L\left[\hat{\bar{\Phi}}_4, \mathcal{W}_1^I, \Phi_{\hat{P}}\right] \frac{1}{s_{14}} A_R\left[\Phi_{-\hat{P}}, \bar{\mathcal{W}}_2^J, \hat{\Phi}_3\right]$$

$$= -\oint_{\{z=0\}} \frac{dz}{z} \frac{z_I}{z - z_I} \frac{\hat{x}_{14}\hat{x}_{23}[\hat{P}1^I][\hat{P}2^J]}{m_1 m_2\langle q\hat{P}\rangle^2} \frac{1}{s_{14}} \int d^2\eta_{\hat{P}} \delta^{(2)}\left(\hat{Q}_L\right) \delta\left(\langle q\hat{Q}_L^{\dagger}\rangle\right) \delta^{(2)}\left(\hat{Q}_R\right) \delta\left(\langle q\hat{Q}_R^{\dagger}\rangle\right)$$

$$= -\oint_{\{z=0\}} \frac{dz}{z} \frac{z_I}{z - z_I} \frac{\hat{x}_{23}[\hat{P}1^I][\hat{P}2^J]}{m_1 m_2} \frac{1}{s_{14}} \left(1 + \frac{\hat{x}_{23}}{\hat{x}_{14}}\right)^{-1} \delta^{(2)}\left(Q^{\dagger}\right) \delta^{(2)}(Q)$$

$$\begin{aligned}
&= -\oint_{\{z=0\}} \frac{dz}{z} \frac{z_I}{z-z_I} \frac{1}{s_{12}s_{14}} [\hat{P}1^I][\hat{P}2^J] (\hat{x}_{14} + \hat{x}_{23}) \delta^{(2)}(Q^\dagger) \delta^{(2)}(Q) \\
&= -\oint_{\{z=0\}} \frac{dz}{z} \frac{z_I}{z-z_I} \frac{1}{s_{12}s_{14}} \frac{[\hat{P}1^I][\hat{P}2^J]}{[\rho\hat{P}]} \left( \frac{[\rho|p_1|\hat{P}\rangle}{m_1} + \frac{[\rho|p_2|\hat{P}\rangle}{m_2} \right) \delta^{(2)}(Q^\dagger) \delta^{(2)}(Q) \\
&= -\oint_{\{z=0\}} \frac{dz}{z} \frac{z_I}{z-z_I} \frac{1}{s_{12}s_{14}} \left( \langle 1^I|\hat{P}|2^J] + [\hat{1}^I|\hat{P}|2^J\rangle \right) \delta^{(2)}(Q^\dagger) \delta^{(2)}(Q) \\
&= -\oint_{\{z=0\}} \frac{dz}{z} \frac{z_I}{z-z_I} \frac{1}{s_{12}s_{14}} \left( \langle 1^I|\hat{p}_4|2^J] + [1^I|\hat{p}_4|2^J\rangle + m_4[1^I 2^J] + m_4\langle 1^I 2^J\rangle \right) \delta^{(2)}(Q^\dagger) \delta^{(2)}(Q) \\
&= \left[ \langle 1^I|p_4|2^J] + [1^I|p_4|2^J\rangle + m_4[1^I 2^J] + m_4\langle 1^I 2^J\rangle + z_I\langle 1^I|r|2^J] + z_I[1^I|r|2^J\rangle + \mathcal{B} \right] \\
&\qquad \times \frac{\delta^{(2)}(Q^\dagger) \delta^{(2)}(Q)}{s_{12}s_{14}}.
\end{aligned} \tag{147}$$

To obtain the fourth equality from the third we have used Eq.(142). In the last equality we have computed residue at $z = z_I$ and along with the pole at infinity. To evaluate the boundary term, $\mathcal{B}$, we substitute $z = \frac{1}{u}$ and calculate the residue around $u = 0$,

$$\begin{aligned}
\mathcal{B} &= -\oint_{u=0} \frac{du}{u} \frac{z_I}{1 - z_I u} \left( \langle 1^I|r|2^J] + [1^I|r|2^J\rangle \right) \\
&= -z_I \left( \langle 1^I|r|2^J] + [1^I|r|2^J\rangle \right).
\end{aligned} \tag{148}$$

Therefore the required four-point amplitude is,

$$A_4\left[\mathcal{W}_1^I, \bar{\mathcal{W}}_2^J, \Phi_3, \bar{\Phi}_4\right] = \left[ \langle 1^I|p_4|2^J] + [1^I|p_4|2^J\rangle + m_4[1^I 2^J] + m_4\langle 1^I 2^J\rangle \right] \frac{\delta^{(2)}(Q^\dagger) \delta^{(2)}(Q)}{s_{12}s_{14}}. \tag{149}$$

We note that even though the amplitude matches with the $\mathcal{N} = 2^*$ amplitude, the channel we have considered here does not exist for $\mathcal{N} = 2^*$ theory since we have used massless hypermultiplet exchange. However, it would exist in a theory where massless hypermultiplets are coupled to massive $\mathcal{N} = 2$ SYM and hypermultiplets. Therefore the above calculation applies for such a theory.

## D.2  4-point hypermultiplet

We want to evaluate the four-point amplitude $A_4\left(\Phi_1, \bar{\Phi}_2, \Phi_3, \bar{\Phi}_4\right)$ with massless spin-1 exchange. We consider shifts in legs 1 and 3, given by,

$$\hat{p}_1 = p_1 zr, \qquad \hat{p}_3 = p_3 - zr. \tag{150}$$

The amplitude is then obtained by summing over factorization channels, $u = -\left(p_1 + p_4\right)^2$ and $s = -\left(p_1 + p_2\right)^2$.

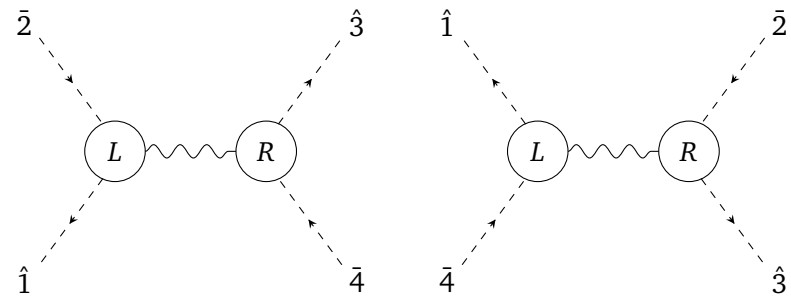

BCFW analysis yields,

$$A_4\left(\Phi_1,\bar{\Phi}_2,\Phi_3,\bar{\Phi}_4\right) = -\oint_{\{z=0\}} \frac{\mathrm{d}z}{z}\frac{z_{I,1}}{z-z_{I,1}}\sum_{h=\pm}\int \mathrm{d}^2\eta_{\hat{P}} A_L\left(\bar{\Phi}_4,\hat{\Phi}_1,G_{\hat{P}}^h\right)\frac{-1}{s_{14}}A_R\left(G_{-\hat{P}}^{-h},\bar{\Phi}_2,\hat{\Phi}_3\right)$$

$$-\oint_{\{z=0\}} \frac{\mathrm{d}z}{z}\frac{z_{I,2}}{z-z_{I,2}}\sum_{h=\pm}\int \mathrm{d}^2\eta_{\hat{P}} A_L\left(\hat{\Phi}_1,\bar{\Phi}_2,G_{\hat{P}}^h\right)\frac{-1}{s_{12}}A_R\left(G_{-\hat{P}}^{-h},\hat{\Phi}_3,\bar{\Phi}_4,\right),\quad (151)$$

with $z_{I,1} = \frac{s_{14}}{2r\cdot(p_1+p_4)}$ and $z_{I,2} = \frac{s_{12}}{2r\cdot(p_1+p_2)}$ are the location of simple poles when the two propagators go on-shell respectively.

Let us first consider the factorization on u-channel. We have,

$$A_L\left(\bar{\Phi}_4,\hat{\Phi}_1,G_{\hat{P}}^+\right)A_R\left(G_{-\hat{P}}^-,\bar{\Phi}_2,\hat{\Phi}_3\right) = \frac{1}{\langle q\hat{P}\rangle\langle q|p_2|\hat{P}]}\delta^{(2)}\left(\hat{Q}_L\right)\delta\left(\langle q\hat{Q}_L^\dagger\rangle\right)\delta^{(2)}\left(\hat{Q}_R^\dagger\right)\delta\left(\langle q|p_2|\hat{Q}_R]\right)$$

$$= \frac{\hat{x}_{23}}{m_2\langle q\hat{P}\rangle^2}\delta^{(2)}\left(\hat{Q}_L\right)\delta\left(\langle q\hat{Q}_L^\dagger\rangle\right)\delta^{(2)}\left(\hat{Q}_R^\dagger\right)\delta\left(\langle q|p_2|\hat{Q}_R]\right),$$

$$A_L\left(\bar{\Phi}_4,\hat{\Phi}_1,G_{\hat{P}}^-\right)A_R\left(G_{-\hat{P}}^+,\bar{\Phi}_2,\hat{\Phi}_3\right) = \frac{1}{\langle q|\hat{p}_1|\hat{P}]\langle q\hat{P}\rangle}\delta^{(2)}\left(\hat{Q}_L^\dagger\right)\delta\left(\langle q|\hat{p}_1|\hat{Q}_L]\right)\delta^{(2)}\left(\hat{Q}_R\right)\delta\left(\langle q\hat{Q}_R^\dagger\rangle\right)$$

$$= \frac{\hat{x}_{14}}{m_1\langle q\hat{P}\rangle^2}\delta^{(2)}\left(\hat{Q}_L^\dagger\right)\delta\left(\langle q|\hat{p}_1|\hat{Q}_L]\right)\delta^{(2)}\left(\hat{Q}_R\right)\delta\left(\langle q\hat{Q}_R^\dagger\rangle\right),$$

$$(152)$$

where,

$$\hat{x}_{14} = \frac{[\rho|\hat{p}_1|\hat{P}\rangle}{m_1[\rho\hat{P}]} = \frac{m_1\langle q\hat{P}\rangle}{\langle q|\hat{p}_1|\hat{P}]}, \qquad \hat{x}_{23} = \frac{[\rho|p_2|\hat{P}\rangle}{m_2[\rho\hat{P}]} = \frac{m_2\langle q\hat{P}\rangle}{\langle q|p_2|\hat{P}]}. \qquad (153)$$

Supersymmetric charges are expressed as,

$$\begin{aligned}
\hat{Q}_L &= |\hat{1}^I]\eta_{\hat{1},I} - |4^I]\eta_{4,I} + |\hat{P}]\tilde{\eta}_{\hat{P}}^\dagger, \\
\hat{Q}_L^\dagger &= -|\hat{1}^I\rangle\eta_{\hat{1},I} - |4^I\rangle\eta_{4,I} + |\hat{P}\rangle\eta_{\hat{P}}, \\
\hat{Q}_R &= -|\hat{P}]\tilde{\eta}_{\hat{P}}^\dagger - |2^I]\eta_{2,I} + |\hat{3}^I]\eta_{\hat{3},I}, \\
\hat{Q}_R^\dagger &= -|\hat{P}\rangle\eta_{\hat{P}} - |2^I\rangle\eta_{2,I} - |\hat{3}^I\rangle\eta_{\hat{3},I}.
\end{aligned} \qquad (154)$$

It can be checked that,

$$\begin{aligned}
\langle\hat{P}|\hat{p}_1|\hat{Q}_L] &= m_1\langle\hat{P}\hat{Q}_L^\dagger\rangle, \\
\langle\hat{P}|p_2|\hat{Q}_R] &= -m_2\langle\hat{P}\hat{Q}_R^\dagger\rangle.
\end{aligned} \qquad (155)$$

We also note that,

$$\delta^{(2)}\left(\hat{Q}_L\right)\delta\left(\langle q\hat{Q}_L^\dagger\rangle\right) = \frac{1}{m_1}\delta^{(2)}\left(\hat{Q}_L^\dagger\right)\delta\left(\langle q|\hat{p}_1|\hat{Q}_L]\right). \qquad (156)$$

and similarly for right delta function.

Now, using Eq.(152) and summing over both helicities in the exchange we get,

$$\sum_{h=\pm} A_L\left(\bar{\Phi}_4,\hat{\Phi}_1,G_{\hat{P}}^h\right)A_R\left(G_{-\hat{P}}^{-h},\bar{\Phi}_2,\hat{\Phi}_3\right)$$

$$= \frac{\left(\hat{x}_{23}+\hat{x}_{14}\right)}{m_1 m_2\langle q\hat{P}\rangle^2}\delta^{(2)}\left(\hat{Q}_L^\dagger\right)\delta\left(\langle q|\hat{p}_1|\hat{Q}_L]\right)\delta^{(2)}\left(\hat{Q}_R^\dagger\right)\delta\left(\langle q|p_2|\hat{Q}_R]\right)$$

$$= \frac{\left(\hat{x}_{23}+\hat{x}_{14}\right)}{m_1 m_2\langle q\hat{P}\rangle^2}m_2\left(1+\frac{\hat{x}_{23}}{\hat{x}_{14}}\right)^{-1}\delta^{(2)}\left(Q^\dagger\right)\delta^{(2)}\left(Q\right)\delta\left(\langle q|\hat{p}_1|\hat{Q}_R]\right)\delta\left(\langle q\hat{Q}_R^\dagger\rangle\right). \qquad (157)$$

Performing the $\eta_{\hat{P}}$ integral we get,

$$\int d^2 \eta_{\hat{P}} \, \delta \left( \langle q | \hat{p}_1 | \hat{Q}_R ] \right) \delta \left( \langle q \hat{Q}_R^\dagger ] \right) = \langle q | \hat{p}_1 \hat{P} | q \rangle = m_1 \frac{\langle q \hat{P} \rangle^2}{\hat{x}_{14}} \,. \tag{158}$$

Therefore contribution from the u-channel is,

$$\frac{1}{s_{14}} \delta^{(2)} \left( Q^\dagger \right) \delta^{(2)} (Q) \,. \tag{159}$$

Similar contribution comes from s-channel where $s_{14}$ is replaced by $s_{12}$.

Using $s_{12} + s_{14} = s_{13}$, the four-point amplitude is determined to be,

$$A_4 \left( \Phi_1, \bar{\Phi}_2, \Phi_3, \bar{\Phi}_4 \right) = \frac{s_{13}}{s_{14} s_{12}} \delta^{(2)} \left( Q^\dagger \right) \delta^{(2)} (Q) \,. \tag{160}$$

We note that since we have used the massless $\mathcal{N} = 2$ SYM here in the intermediate leg, this amplitude is appropriate for the origin of the moduli space of $\mathcal{N} = 2^*$ theory.

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
