# Peer review of "Scattering Amplitudes and BCFW in $\mathcal{N}=2^*$ Theory"

_SciPost Physics, doi:SciPost Phys. 13, 008 (2022)_

## Round 1 · Referee Report · Anonymous (Referee 2) · 2022-4-29

Report
The analysis in the paper highlights important structural differences between BCFW recursion in N=4 and N=2* theory. In particular, the appearance of little-group noncovariant terms in
intermediate steps of the computation. The cancellation of these terms in obtaining the final answer is an important consistency check of the method.
The paper is largely well written and the main results are presented clearly. Expectedly, these results would pave the way for a more exhaustive exploration of scattering amplitudes
in the moduli space of N=2* theories and more generally, N=2 theories.
We recommend the manuscript for publication, pending the addressal of a few comments, given below.
Requested changes
-
The authors may clarify if they are working with color ordered amplitudes.
-
The introduction could also be slightly expanded to provide general motivations for the study of N=2* theories, including further references to the literature if needed.
-
Some terms could also be clearly defined when used for the first time, e.g. a. 'u-spinor' on page 3 b. 'band structure' in Section 3.4
-
On page 23, " In N = 4 SYM Coulomb branch, it was found that.." should have the relevant reference.
-
Some sentences could also be editorialized. For example: a. In the Introduction: The sentence beginning with "We put long in quotes because.." b. In the Conclusions: "Also, because the pole at infinity recursive structure of the amplitudes is nontrivial,"

---

## Round 1 · Referee Report · Anonymous (Referee 1) · 2022-4-29

Strengths
It contains mostly the essential references which makes it simpler to follow
Weaknesses
Report
In this paper the authors have studied tree level scattering amplitudes (in particular they have explicitly computed three and four point amplitudes) for $\mathcal{N} = 2^*$ super Yang-Mills (SYM) theory in the Coloumb branch, building up on works [arXiv : 1902.07204 & arXiv : 1902.07205] by Herderschee, Koren and Trott. The authors have used massive spinor-helicity variables and Britto-Cachazo-Feng-Witten (BCFW) formalism to compute the amplitudes. More preciously they use following two different methods.
1. Method of projection from $\mathcal{N} = 4$ SYM : Using this method three and four-point amplitudes have been computed in section 3 and section 4 of the paper. These amplitudes are the main results of this work. The three point amplitudes are also written in terms of $u$-spinors such that they can be conveniently used to build up BCFW recursion in section 5.
2. Massive super-BCFW recursion relation : In section 5, four point amplitudes are re-derived using $u$-spinor representations of the three point amplitudes mentioned above. This BCFW analysis is structurally different from $\mathcal{N} = 4$ SYM (although the method mentioned above shows the amplitudes can be obtained as projection from $\mathcal{N} = 4$ SYM), due to presence of 'pole at $\infty$'. In more detail, the massive BCFW shifts are $not$ covariant under the little group e.g. the integrand contains explicit BCFW shift parameter ($z$, in their notation). But for the particular example the authors are studying, they find that this 'non-covariance' is precisely canceled by the pole at $z = \infty$. This is not unexpected because the only way the recursion could work is when the two troublesome contributions to the amplitudes cancel each other. But this feature gives some technical advantages in the computation, since one can ignore both the contributions ($z$-dependent piece in the integrand and the residue at $z = \infty$) from the beginning.
The manuscript is technically very well written and it extends the applications of on-shell methods in computing tree-level amplitudes to larger class of supersymmetric theories. I recommend the manuscript for publication in SciPost Physics.
Requested changes
Here are some suggestions that, I believe, should improve the manuscript and make it more accessible to the readers - particularly working broadly in scattering amplitudes related topics.
I feel the paper lacks proper motivations. After briefly reviewing the success of on-shell amplitude program in $\mathcal{N} = 4$ SYM, it somewhat abruptly starts discussing about $\mathcal{N} = 2^*$ theory (in the third paragraph of the introduction) as : "Another place where this generalised spinor helicity formalism can be used is in studying amplitudes in the $\mathcal{N} = 2^*$ theory ...". The authors may consider adding a paragraph there explaining why one should be interested in studying $\mathcal{N} = 2^*$ SYM amplitudes in the first place (e.g. describing if there are some interesting issues/results that can be tackled/checked by studying these amplitudes). In the beginning of section 5, possible connections with massive amplitudes in $\mathcal{N} = 2$ SYM have been mentioned. This part may also be expanded in the introduction.

---

## Round 2 · Referee Report · Anonymous · 2022-5-24

Report

I am happy with the changes made by the authors in the introduction.

---

## Round 2 · Referee Report · Anonymous · 2022-5-24

Report

We thank the authors for incorporating changes in the draft. We recommend publication of the draft in its current form.

---

## Round 2 · Author Response

We thank the referees for their comments which helped improve the manuscript. We have incorporated all the suggestions of referees and the changes in the manuscript are in red color for the ease of perusal for the referee. Nevertheless we will briefly summarize changes in the manuscript below.

  1. We have made four changes in the introduction.

  2. We have added text in subsections 3.2, 3.3, and 3.4.

  3. In section 4 we have emphasized that the amplitudes under consideration are color ordered.

  4. Finally we have added a speculation in the conclusion section.

---

## Round 2 · List of Changes

Changes in the introduction are as follows.

1A. We have expanded on our motivation in taking up the N=2* problem, which has intermediate level of complexity compared to N=4 and N=2. We have also added a couple of references in this amendment.

1B. We reiterate this point again in the context of the BCFW relations.

1C. We have rephrased the sentence about the long multiplet.

1D. We have elaborated a bit on the u-spinor and pointed the reader to section 3 for more details.

2A. In subsection 3.2, we explicitly mention that we are looking at the color ordered amplitudes.

2B. In subsection 3.3, we elaborate on the massive fields by using the example of excursion in the Coulomb branch.

2C. In subsection 3.4, we add a preamble to the section by mentioning about the band structure in the N=4 SYM theory.

3. We added a parenthetical remark on color ordering in the first line of section 4.

4. In the Conclusion section, we speculate about the BCFW shift for higher point functions for massive fields.

---

## Editorial Decision

published